# Porcine B cell receptor repertoire uncovers balanced recognition of antigenic structures on serotype Asia1 foot-and-mouth disease virus

Shulun Huang[1,2,3], Shanquan Wu[4], Fengjuan Li[1,3], Pinghua Li[1,3], Pu Sun[1,3], Yimei Cao[1,3], Huifang Bao[1,3], Kaiheng Dong[1,3], Jiaxin Yang[1,3], Hehe Zhang[1,3], Qiongqiong Zhao[1,3], Ying Sun[1,3], Dong Li[1,3], Xingwen Bai[1,3], Yuanfang Fu[1,3], Hong Yuan[1,3], Xueqing Ma[1,3], Zhixun Zhao[1,3], Jing Zhang[1,3], Jian Wang[1,3], Zaixin Liu[1,3], Yong Peng[5], Kun Li[1,3]*, Jinlian Hua[2]*, Zengjun Lu [1,3]*, Dongsheng Lei[1,4,6]*, Qiang Zhang[1,3]*

1 State Key Laboratory of Animal Disease Control and Prevention, College of Veterinary Medicine, Lanzhou University, Lanzhou Veterinary Research Institute, Chinese Academy of Agricultural Sciences, Lanzhou, China, 2 College of Veterinary Medicine, Northwest A&F University, Shaanxi Centre of Stem Cells Engineering & Technology, Yangling, Shaanxi, China, 3 Key Laboratory of Magnetism and Magnetic Functional Materials, School of Physical Science and Technology, Electron Microscopy Centre of Lanzhou University, Lanzhou University, Lanzhou, China, 4 Gansu Province Research Center for Basic Disciplines of Pathogen Biology, Lanzhou, China, 5 School of Materials and Energy, Electron Microscopy Centre of Lanzhou University, Lanzhou University, Lanzhou, China, 6 Jiangsu Key Laboratory of Zoonosis, Yangzhou University, Yangzhou, China

* likun02@caas.cn (KL); jinlianhua@nwsuaf.edu.cn (JH); luzengjun@caas.cn (ZL); leids@lzu.edu.cn (DL); zhangqiang@caas.cn (QZ)

## Abstract

Of the seven serotypes of foot-and-mouth disease virus (FMDV) strains circulating globally, serotype Asia1 has been effectively eradicated in China through systematic vaccination in livestock. The structural characteristics of serotype Asia1 may enhance its immunogenicity compared to other serotypes. Herein, we present a preliminary exploration of Asia1-binding B-cell receptor repertoire, containing 3571 clones, and identified 17 porcine-derived neutralizing monoclonal antibodies (pnAbs) from the top 33 high-frequency clonotypes. The majority of pnAbs (14/17) recognized the epitopes on VP2, with a common determinant at residue 72 (D) on the B-C loop; two pnAbs (2/17) recognized a novel epitope spanning VP2 and VP3; and the remaining one (1/17) bound to the C-terminus of VP1. Furthermore, the antigenic structures on VP2 and spanning VP2 and VP3 were respectively elucidated by determining the cryo-EM structures of FMDV serotype Asia1 in complexes with two pnAbs, PAS5 and PAS12. The light chain of PAS5, forming the majority of contact sites with the viral particle, focuses on the βB, B-C loop, βC and H-I loop of VP2, with key determinants at residues 68, 72 and 77 around the three-fold axis, corresponding to antigenic site 2. The contact sites of both $V_H$ and $V_L$ of PAS12 uncover a novel antigenic structure comprising the B-C, and H-I loops on VP2, and the B-B knob and βB on VP3, with key determinants at residue 73 on VP2 and 59 on VP3. Subsequently, site-directed competitive ELISA analysis of sera from primary and booster vaccinated pigs revealed a

**Data availability statement:** The scBCR-seq and scRNA-seq data generated in this study have been deposited in the Sequence Read Archive database with the accession code of PRJNA1203304. The cryo-EM density maps and structures for FMDV-Asia1-PAS5 and FMDV-Asia1-PAS12 have been deposited at the Electron Microscopy Data Bank (EMDB) and the Protein Data Bank (PDB) with the following accession numbers: FMDV-Asia1-PAS5, EMD-48509, PDB 9MQ0, and FMDV-Asia1-PAS12, EMD-48508, PDB 9MPZ.

**Funding:** This work was supported by grants from the National Key R&D Program of China (2021YFD1800304 to Z.L.), the National Natural Science Foundation of China (Nos. 32373028 to K.L., 32072873 to Y.C., 32171300 to D.L.), the Innovation Program of Chinese Academy of Agricultural Sciences (CAAS-CSLPDCP-202402 to Z.L.), and Fundamental Research Funds for the Central Universities (lzujbky-2021-ct05 to D.L.). The funders had no role in study design, data collection and analysis, decision to publish, or preparation of the manuscript.

**Competing interests:** The authors have declared that no competing interests exist.

balanced antibody response profile, suggesting a potentially even immunodominance among antigenic site 2, VP1 G-H loop, and the novel antigenic structure spanning VP2 and VP3 on FMDV serotype Asia1. Compared to the focused immunodominance observed in other serotypes, this balanced antigenic recognition across VP1, VP2, and VP3 of FMDV serotype Asia1 reflects a diversified antibody response that may contribute to effective neutralization and protection.

## Author summary

Foot-and-mouth disease virus serotype Asia1, having good immunogenicity, was effectively eradicated in China through livestock vaccination. However, the antigenic properties of Asia1 remain unknown. We constructed a porcine B-cell receptor repertoire against FMDV serotype Asia1 and identified a panel of neutralizing antibodies from high-frequency clonotypes. Furthermore, the antigenic structures on VP2 and spanning VP2 and VP3 of serotype Asia1 were elucidated from cryo-EM complexes of virus-antibodies. Next, by evaluating competitive antibodies in sera from vaccinated pigs, we observed that FMDV serotype Asia1 exhibits a more balanced antibody response profile across its structural proteins. The evenly distributed immunodominance of antigenic structures on viral particle reflects a diversified antibody response, which may contribute to the effective immunogenicity of FMDV serotype Asia1. This configuration of antigenic structures is beneficial for guiding rational vaccine design.

## Introduction

Foot-and-mouth disease virus (FMDV) is an extremely contagious pathogen affecting livestock and other cloven-hoofed animals, existing as seven immunologically distinct serotypes, namely, A, O, C, Asia1, SAT1, SAT2 and SAT3 [1]. These serotypes have been further classified into genotypes or topotypes based on VP1 nucleotide sequence divergence exceeding 15% [2,3]. Among these, serotypes O and A are still the most prevalent, whereas serotype Asia1 has not been detected since 2009 and was declared eradicated in 2019 due to the compulsory immunization program in China [4]. FMDV serotype Asia1 was first identified in 1951 and restricted to certain regions of Asia [5,6]. Isolates of FMDV serotype Asia1 have been categorized into nine different groups (GI~GIX) within a single topotype, based on approximately 5% divergence in the VP1 coding sequence [7–11]. Serotype Asia1 is considered to be the least genetically and antigenically diverse serotype, and its structural properties may enhance immunogenicity compared to other serotypes. To date, the antigenic structures of FMDV serotypes A and O have been extensively characterized by resolving cryo-EM complex structures of virus-antibody [12–17]. However, the structural information regarding FMDV serotype Asia1 remains unknown. Given that it was

the first eradicated serotype through vaccination of livestock, elucidating its antigenic characteristics could provide insights for the design of effective vaccines to prevent other serotypes of FMDV.

Neutralizing monoclonal antibodies (nAbs) are crucial tools for dissecting the antigenic structure of FMDV and for understanding viral evolution under immune pressure. Currently, the identification of antigenic sites on serotype Asia1 has mainly been based on murine nAbs [18,19]. Site 1 is mainly formed by critical residues 140–142 upstream of the conserved RGD motif in the G-H loop of VP1, whereas key epitopes at the C-terminus corresponding to other serotypes have not been reported. Site 2 is defined by substitutions at residues 67, 72, 74, 77 and 79 on the VP2 B-C loop. Site 4 involves residues 58 and 59 on the VP3 B-B knob. Additionally, an epitope involving a critical residue 218 at the VP3 C-terminus has been documented in this serotype [18].

The diversity of antibody repertoires is generated through V(D)J germline gene segment recombination, and the number of these segments varies substantially among mammalian species. Germline V genes of the heavy chain ($V_H$) can be divided into three main clades, all of which are present in humans and mice, whereas porcine $V_H$ genes are exclusively restricted to clade III [20]. Mice possess more than 100 $V_H$ genes, while pigs and cattle have fewer than 30 [20,21]. Two distinct types of light chains (kappa and lambda) are present in the mammalian immune system. Adult swine exhibit a balanced expression of kappa and lambda chains, similar to humans but distinct from rodents and cattle, which preferentially utilize kappa and lambda chains, respectively [21,22]. In addition to these genetic distinctions, pigs display unique pathological features following FMDV infection. Unlike ruminants, pigs develop more severe clinical disease, marked by high viral excretion during the acute phase and efficient viral clearance without progression to a carrier state [23]. These species-specific differences in viral pathogenesis are accompanied by distinct B cell repertoire composition and immune response kinetics, which may influence antibody maturation pathways and epitope recognition. Consequently, porcine-derived nAbs may serve as more suitable tools than those derived from mice, as they can more accurately reflect the immune response to viral antigenic structure in vivo. High-throughput B-cell receptor (BCR) repertoire analysis from the natural host is therefore essential for comprehensive antigenic profiling and for identifying nAbs targeting FMDV serotype Asia1.

The Asia1/JS/05 strain, belonging to the GV genotype of FMDV serotype Asia1, was isolated from Jiangsu province in China [24]. Due to its strong immunogenicity, this strain was used as a vaccine strain for effective control of viral spread in China. Here, we constructed an antigen-binding BCR repertoire of the Asia1/JS/05 strain from a vaccinated pig. A total of 17 porcine-derived neutralization antibodies (pnAbs) were identified from 33 high-frequency clonotypes and mapped to three distinct antigenic sites on the surface of FMDV serotype Asia1. Furthermore, the cryo-EM complex structures of FMDV serotype Asia1 with two pnAbs were resolved, revealing two antigenic structures on VP2 and across VP2 and VP3. Unlike serotypes O and A, FMDV serotype Asia1 presented an evenly distributed immunodominance of antigenic structures on VP1, VP2 and VP3, which could contribute to exceptional immunogenicity of the vaccine molecule.

## Results

### Phenotypes, composition, and germline gene usage of the porcine B-cell repertoire against FMDV serotype Asia1

To characterize the B-cell response landscape against FMDV serotype Asia1 in a vaccinated pig, we developed a protocol to isolate antigen-binding B cells from porcine peripheral blood mononuclear cells (PBMCs) using the Asia1/JS/05 strain as bait. We then constructed a single B-cell repertoire using high-throughput single-cell BCR and transcriptome sequencing (Fig 1A and Fig A in S1 Text). Serum samples collected at various time points post-vaccination were evaluated using virus neutralization test (VNT) against the Asia1/JS/05 strain. The data showed that booster immunization with FMDV serotype Asia1 induced robust neutralizing antibody responses against the homologous strain (titers > 2.71 $\log_{10}$) (Fig 1B). Flow cytometry analysis revealed that Asia1/JS/05-binding B-cell subsets accounted for approximately 0.38% of porcine PBMCs at 5 days post-final immunization, including naive B cells (IgM isotype) and class-switched isotypes (Fig 1C–1E). Following the depletion of T cells, CD14+ monocytes, NK cells and naïve IgM+ B cells from PBMCs via negative magnetic

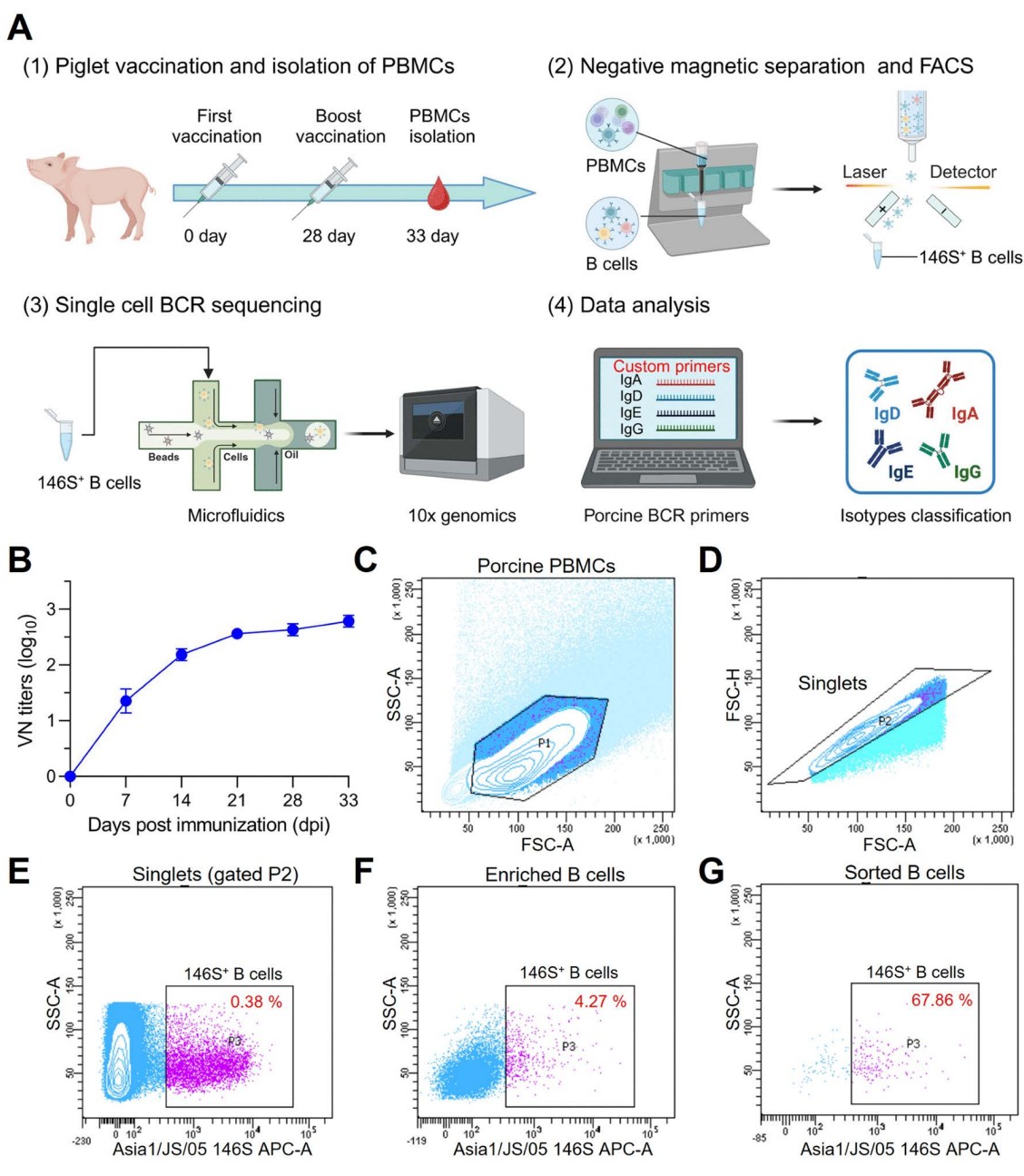

**Fig 1. Construction of FMDV serotype Asia1-binding B-cell repertoire. (A)** Schematic diagram of the immunization schedule and workflow for constructing the FMDV Asia1-binding B cell repertoire. PBMCs collected after secondary vaccination were subjected to negative magnetic separation and fluorescence-activated cell sorting (FACS) to isolate Asia1/JS/05-binding B cells. Subsequently, enriched antigen-binding B cells were processed for single-cell BCR and transcriptome sequencing using the 10 × genomics platform with customized porcine BCR primers. Created in BioRender. Chen, Y. (2026) https://BioRender.com/qnwnmh1. **(B)** Dynamics of serum neutralizing antibody titers against the FMDV Asia1/JS/05 strain at different time points post-vaccination. **(C-G)** Flow cytometric identification of FMDV-binding B cells using biotinylated Asia1/JS/05 146S antigen. **(C)** PBMCs were gated (P1) to exclude cell debris based on lower values of FSC-A and SSC-A. **(D)** Singlets were selected in gate P2 based on the diagonal streak in the FSC-A and FSC-H plot. **(E-G)** Representative plots showing 146S⁺ B cells among porcine PBMCs **(E)**, enriched B cells **(F)**, and sorted B cells (G) were identified in gate P3.

separation, the frequency of antigen-binding IgM⁻ B cells increased to 4.27% (Fig 1F), representing an > 11-fold enrichment compared to unenriched PBMCs (Fig 1E). Next, Asia1/JS/05-binding B cells were isolated from the enriched fraction using fluorescence-activated cell sorting (FACS), achieving a purity of 67.86% (Fig 1G).These purified B cells were then subjected to single-cell V(D)J sequencing using the 10×genomics platform, utilizing porcine-specific primers as previously described [17]. Finally, we successfully established the porcine BCR repertoire binding to FMDV serotype Asia1.

First, we characterized the phenotype of Asia1-binding B-cell repertoire. Uniform manifold approximation and projection (UMAP) analysis revealed that the antigen-binding B cells consisted of predominantly of memory B cells (89.07%), with a small proportion of plasmablasts (PBs, 10.93%) (Fig 2A). Significant upregulation of genes (avg_log$_2$FC > 2) of marker genes, including the joining chain of multimeric IgA and IgM (JCHAIN), marginal zone B and B1 cell specific protein 1 (MZB1) and X-box binding protein 1 (XBP1), was observed in porcine PBs (Fig 2B). The predominance of memory B cells in the B cell repertoire suggested the importance of immunologic memory in porcine humoral response to FMDV serotype Asia1.

Next, we analyzed the antibody features of the BCR repertoire. As shown in Fig 2C, the Asia1-binding BCR repertoire comprised a total of 3571 paired variable regions of heavy chain and light chain ($V_H$ and $V_L$). Isotype analysis showed that the IgG dominated response, accounting for approximately 60.76% of all isotypes. The IgD and IgA isotypes constituted approximately 22.57% and 16.67% of the repertoire, respectively. Regarding light chain usage, the proportion of kappa chains (62.95%) was higher than that of lambda chains (37.05%) in the Asia1-binding BCR library. This distinct usage pattern contrasts with serotypes O- and A- binding BCR repertoires, where low ratio of kappa has been observed [17].

Antibody affinity maturation is mediated by the accumulation of somatic hypermutation (SHM) and class switch recombination (CSR) of immunoglobulin (Ig) genes after B cells encounter antigen. In the Asia1-binding repertoire, the average SHM across all isotypes was approximately 6.09%, ranging from 0% to 35% (Fig 2D). The distribution of SHMs showed that the majority of B cells (> 90%) exhibited values below 8.86%, while only a small subset (< 1%), primarily belonging to IgA and IgG isotypes, displayed rates exceeding 11.90%. Among these isotypes, the IgG isotype exhibited the highest SHM with a median of 6.42%, followed by IgA (5.95%) and IgD (5.21%).

Furthermore, the usage of V(D)J germline gene segments of heavy and light chains ($V_H$, Vκ and $V_λ$) in the repertoire was analyzed. As shown in Fig 2E, 15 $V_H$, 9 Vκ, and 8 $V_λ$ gene segments were identified, producing a total of 635 V-D-J combinations of paired heavy and light chains in the Asia1-binding BCR repertoire. The most frequent $V_H$ gene segment was IGHV1–4, followed by IGHV1–10, IGHV1S2, IGHV1S5, and IGHV1–15, together accounting for 86.96% of the repertoire. IGKV1–11, IGKV2–10, IGLV8–13, IGLV8–19, and IGLV3–4 represented the five most frequently utilized Vκ and $V_λ$ gene segments. Analysis of V gene pairing revealed that the most frequent (> 5%) $V_H$-Vκ/$V_λ$ combinations were IGHV1–4-IGKV1–11, IGHV1–10-IGKV1–11, IGHV1–4-IGKV2–10, and IGHV1S2-IGKV1–11 (Fig 2F). Among the four D gene segments identified, IGHD1 and IGHD2 accounted for 90% of the usage in the BCR repertoire. In addition, J genes utilization was highly skewed, with a strong biased toward IGHJ5, IGKJ2, and IGLJ2 among the three heavy chain and five light chain J genes analyzed.

## Identification of porcine neutralizing antibodies from the BCR repertoire reveals three distinct antigenic sites and the dense epitopes clustering on the VP2 B-C loop of FMDV serotype Asia1

To investigate antibody diversity within the Asia1-binding BCR repertoire, the clonotype was defined by a cluster of B cells with identical amino acids (AA) sequence of the heavy chain complementarity-determining region 3 (HCDR3) and the light chain complementarity-determining region 3 (LCDR3). A total of 2674 clonotypes among 3571 paired antibodies were identified, including 35.6% expanded clonotypes (1277) and 64.4% singleton clonotypes (2294). Notably, the Asia1-binding BCR repertoire contained several high-frequency clonotypes, the antibodies of which likely represent the immune response targeting dominant epitopes of FMDV serotype Asia1 (Fig 3A). To identify immunodominant epitopes on FMDV serotype Asia1, we selected and characterized the top 33 high-frequency clones from the repertoire. These 33 porcine monoclonal antibodies (mAbs) were successfully expressed in 293F cells and further purified by Ni-chelating affinity chromatography. The reactivity of these 33 mAbs was tested using indirect immunofluorescence assay (IFA) and

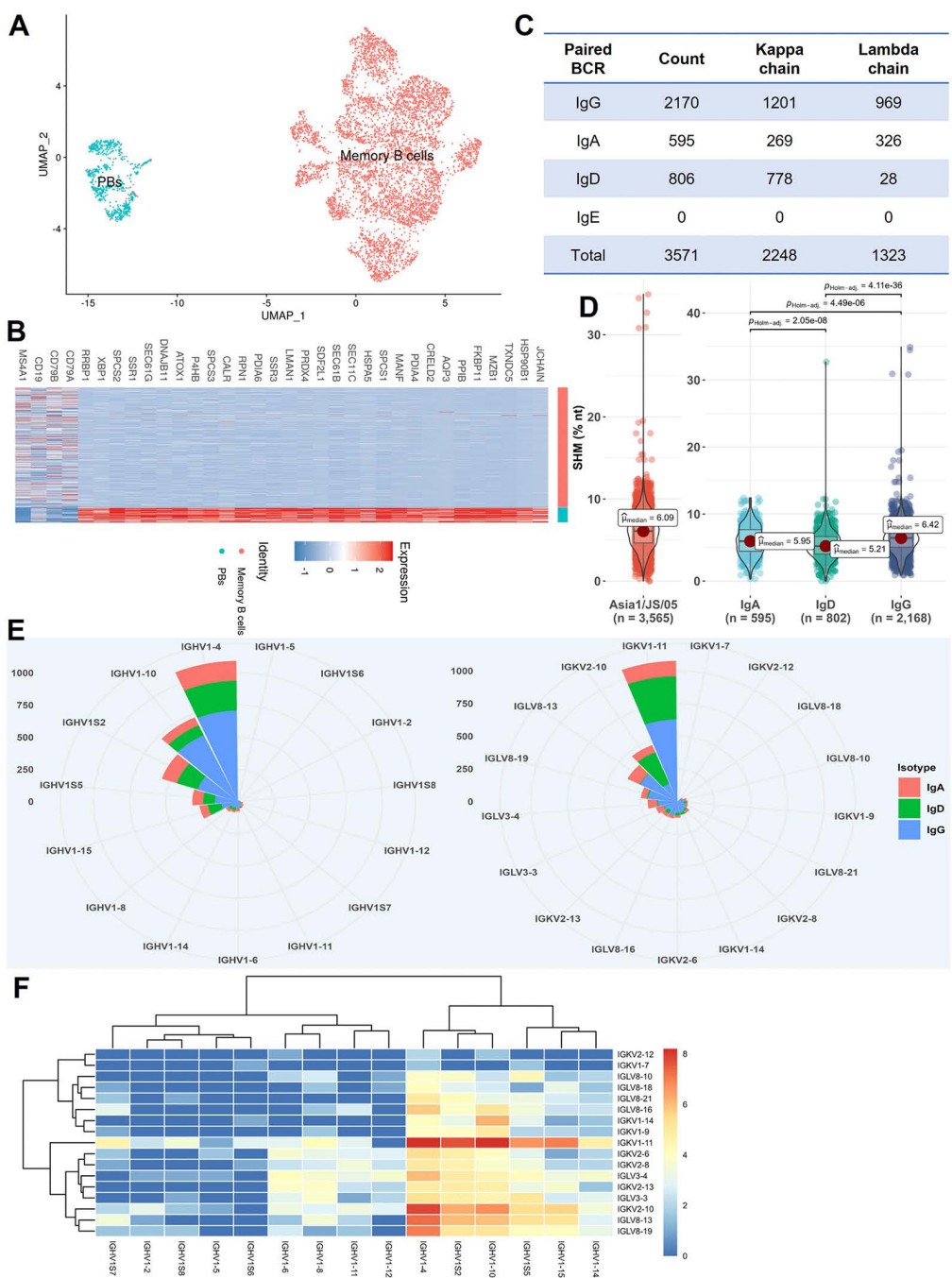

**Fig 2. Molecular characterization of FMDV serotype Asia1 binding B-cell repertoire. (A)** UMAP visualization of single-cell transcriptomic profiles of Asia1/JS/05-binding B cells, showing two major clusters corresponding to PBs and memory B cells. **(B)** Heatmap displaying differentially expressed genes between memory B cells and PBs. **(C)** Overview of clone sizes, antibody isotypes, and light chain types in the porcine BCRs binding to the Asia1/JS/05 strain. **(D)** Comparison of SHM frequencies among different antibody isotypes. Statistical analysis was conducted using the non-parametric Mann-Whitney test in R. P<0.001 indicates an extremely significant difference. **(E)** Frequency proportion of $V_H$ and $V_L$ germline gene usage in the Asia1-binding B-cell repertoire. **(F)** Pairing frequency of $V_H$-$V_L$ germline gene usage in the porcine BCR repertoire targeting FMDV serotype Asia1.

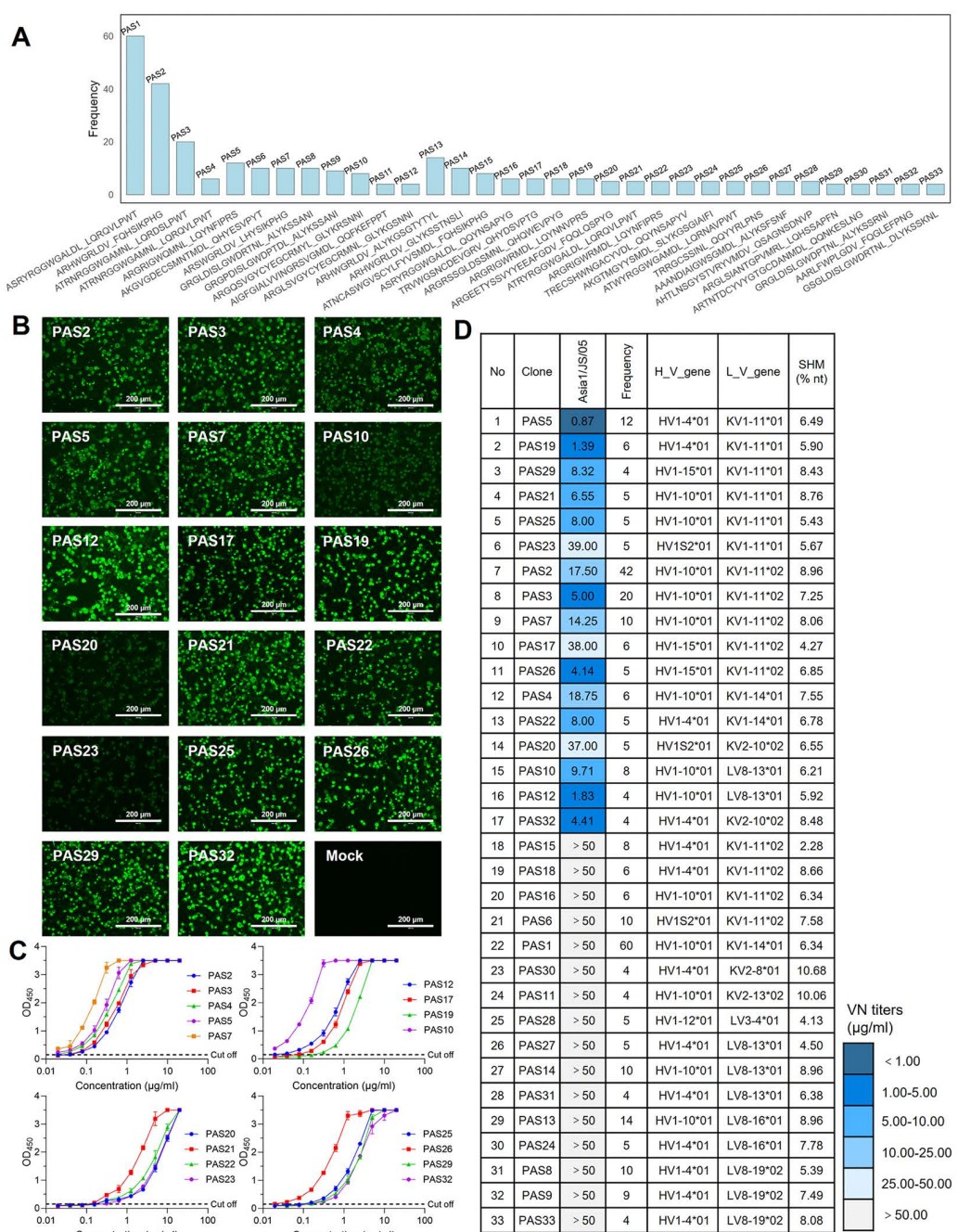

**Fig 3. Characterization of porcine mAbs derived from high-frequency clonotypes in the Asia1/JS/05-binding BCR repertoire. (A)** The counts of the top 33 high-frequency clonotypes in the Asia1-binding porcine BCR repertoire. **(B)** Reactivity of pnAbs against FMDV serotype Asia1 determined by IFA. BHK-21 cells were infected with the Asia1/JS/05 strain and incubated with pnAbs (5 μg/ml), followed by rabbit anti-pig FITC (1: 200 in PBS). Fluorescence signals were observed using an FL Imaging System (Life Technologies, USA). Experiments were independently conducted in triplicate. **(C)** Reactivity of pnAbs against FMDV serotype Asia1 determined using indirect ELISA. **(D)** Neutralization titer, clonotype frequency, H_V gene, L_V gene, and SHM of porcine mAbs derived from high-frequency clonotypes in the Asia1/JS/05-binding repertoire. Neutralization activity against the Asia1/JS/05 strain was assessed by microneutralization assay. Values represent virus neutralization titers (VN) in μg/ml and are shown with a color bar on the right.

enzyme-linked immunosorbent assay (ELISA) against the Asia1/JS/05 strain. As shown in Figs 3B and 3C, and Figs B and C in S1 Text, approximately 90% of these antibodies exhibited different binding activities to the Asia1/JS/05 strain or its whole 146S antigen. These data confirmed that the established porcine antibody repertoire possesses high specificity and reflects the Asia1-specific humoral immune responses in the vaccinated pig. Subsequently, we evaluated the neutralization potency of these mAbs against FMDV serotype Asia1 by VNT. As shown in Fig 3D, half of the mAbs (17/33) were identified as porcine-derived neutralizing antibodies (pnAbs) and exhibited viral neutralization activity against the Asia1/JS/05 isolate.

To explore the antigenic epitopes recognized by pnAbs, neutralization escape mutants were selected using pnAbs against the Asia1/JS/05 strain (Table 1). Based on the sequence analysis of escape mutants and the cross-neutralization efficacy of pnAbs against single-substitution variants, three distinct antigenic sites were identified on FMDV serotype Asia1 (Table 1 and Fig D panels A-C in S1 Text). The majority of pnAbs (14/17) selected for variants carrying substitutions at residues T70, P71, D72, and S74 within the VP2 B-C loop, corresponding to the conventional antigen site 2 described in FMDV serotypes O and A. Notably, all these mutants shared a specific substitution at position 72 (D>N) within VP2, suggesting that this residue is a key determinant of the immunodominant epitopes on VP2 of FMDV serotype Asia1. Two pnAbs, PAS10 and PAS12 (2/17), selected escape variants carrying substitutions at residue E59 on the VP3 B-B knob as well as residues D72 and S74 of the VP2 B-C loop, indicating a novel antigenic structure spanning antigenic sites 2 and 4. On VP1, the C-terminus with critical residues P206 and Q209 was recognized by a single pnAb, PAS32 (1/17).

Notably, the utilization of light-chain V genes exhibited a strong correlation with the mapped epitopes on FMDV serotype Asia1. Specifically, pnAbs recognizing epitopes on VP2 preferentially utilized IGKV1–11/14 gene segments, whereas pnAbs recognizing epitopes spanning VP2 and VP3 were biased toward the IGLV8–13 gene segment (Fig 3D). The pnAb PAS32, which bound to the C-terminus of VP1, and the Asia1-neutralizing antibodies reported to target the VP1 G-H loop both utilized the IGKV2–10 gene segment [17]. This observation may suggest a potential role of the light chain in binding and neutralizing activities of porcine antibodies. Collectively, mapping epitopes using pnAbs revealed the immunodominant nature of the VP2 B-C loop and uncovered a unique antigenic site spanning VP2 and VP3 on FMDV serotype Asia1.

## Immunodominant antigenic structure on VP2 revealed by cryo-EM structure of the FMDV-Asia1-PAS5 complex

To further reveal the immunodominant antigenic structure of FMDV serotype Asia1, we resolved the cryo-EM structure of the single-chain variable fragment (scFv) of PAS5 in complex with Asia1/JS/05 (FMDV-Asia1-PAS5). The cryo-EM reconstruction showed that 60 copies of PAS5-scFv were bound to FMDV-Asia1 capsid around the icosahedral threefold axis (Fig 4A). The final resolution of the cryo-EM reconstruction was determined to be 2.17 Å, based on the gold-standard Fourier shell correlation (FSC) 0.143 cutoff (Fig 4C). The cryo-EM density map was of sufficient quality to enable atomic modeling of most FMDV capsid proteins and the variable loops of the scFv responsible for antigen recognition.

To delineate epitope recognized by PAS5, viral residues within 4 Å of any atom in bound scFv were determined by using the CCP4 software [25]. As shown in FMDV-Asia1-PAS5 complex (Fig 5A–5D), PAS5 contacts the continuous βB, B-C loop, βC and H-I loop on VP2 within one protomer, and forms ten hydrogen bonds and one salt bridge (Tables A and B in S1 Text). The PAS5-interacting residues form a linear epitope on VP2 (Fig E in S1 Text). The antibody-interacting residues in VP2 βB ($_{VP2}$H65, $_{VP2}$L66, $_{VP2}$F67, $_{VP2}$D68), B-C loop ($_{VP2}$T70, $_{VP2}$P71, $_{VP2}$D72, $_{VP2}$L73, $_{VP2}$H77), βC ($_{VP2}$C78, $_{VP2}$H79, $_{VP2}$Y80) and H-I loop ($_{VP2}$V189, $_{VP2}$E195) interact with residues in the three CDRs of light chain and in the FR3 and CDR3 of the heavy chain, namely LCDR3 ($_{VL}$N92, $_{VL}$F93, $_{VL}$I94, $_{VL}$R96), LCDR1 ($_{VL}$S28, $_{VL}$S30, $_{VL}$N32), LCDR2 ($_{VL}$Y50), HCDR3 ($_{VH}$I101) and H-FR3 ($_{VH}$Y59). $_{VP2}$L66 and $_{VP2}$D68 form hydrogen bonds with $_{VL}$N92 and $_{VL}$I94. The side chains of $_{VP2}$T70, $_{VP2}$D72, and $_{VP2}$H77 form hydrogen bond with $_{VL}$R96, $_{VH}$I101 and $_{VL}$Y50, respectively. Additionally, $_{VP2}$D72 forms a salt bridge with the side chain of $_{VL}$R96, indicating the importance of this determinant in forming the epitope.

These interactions of PAS5 with the intact 146S antigen result in a high binding affinity (KD = 2.32 nM) (Fig 5E). Notably, the PAS5 light chain establishes the majority of contact sites with the viral particle. Specifically, residues $_{VP2}$H79, $_{VP2}$C78

**Table 1. Porcine neutralizing mAb escape mutants against Asia1/JS/05 strain.**

| pnAb | Parent virus | Frequency of mutants$ | Residue change | Neutralization concentration# (µg/ml) |
|---|---|---|---|---|
| PAS2 | Asia1/JS/05 | 6/8 | VP2 D72G | 400 |
| | | 2/8 | VP2 D72E | 400 |
| PAS3 | Asia1/JS/05 | 5/8 | VP2 D72G | 400 |
| | | 1/8 | VP2 D72A | 400 |
| | | 1/8 | VP2 P71S D72E | 400 |
| | | 1/8 | VP2 P71S D72G | 400 |
| PAS4 | Asia1/JS/05 | 8/8 | VP2 D72G | 400 |
| PAS5 | Asia1/JS/05 | 2/7 | VP2 D72V | 400 |
| | | 1/7 | VP2 D72N | 400 |
| | | 1/7 | VP2 D72G | 400 |
| | | 1/7 | VP2 D72A | 400 |
| | | 1/7 | VP2 D72Y | 400 |
| | | 1/7 | VP2 T70A D72N | 400 |
| PAS7 | Asia1/JS/05 | 8/8 | VP2 D72G | 400 |
| PAS17 | Asia1/JS/05 | 2/5 | VP2 D72G | 400 |
| | | 3/5 | VP2 S74P | 400 |
| PAS19 | Asia1/JS/05 | 2/5 | VP2 D72N | 400 |
| | | 3/5 | VP2 D72V | 400 |
| PAS21 | Asia1/JS/05 | 7/7 | VP2 D72G | 400 |
| PAS22 | Asia1/JS/05 | 2/5 | VP2 D72G | 400 |
| | | 2/5 | VP2 D72E | 400 |
| | | 1/5 | VP2 P71S | 400 |
| PAS25 | Asia1/JS/05 | 4/7 | VP2 D72G | 400 |
| | | 3/7 | VP2 D72E | 400 |
| PAS29 | Asia1/JS/05 | 7/7 | VP2 D72A | 400 |
| PAS10 | Asia1/JS/05 | 2/8 | VP2 D72N S74P;<br>VP3 E59K | 400 |
| | | 3/8 | VP2 D72N S74P;<br>VP3 E59K;<br>VP1 R199H | 400 |
| | | 1/8 | VP2 P71S S74P;<br>VP3 E59K | 400 |
| | | 2/8 | VP2 P71S S74P;<br>VP3 E59K;<br>VP1 R199H | 400 |
| PAS12 | Asia1/JS/05 | 3/8 | VP2 D72N;<br>VP3 E59K | 400 |
| | | 3/8 | VP2 D72N S74P M95T;<br>VP3 E59G | 400 |
| | | 1/8 | VP2 D72N S74T M95T;<br>VP3 E59G | 400 |
| | | 1/8 | VP2 D72G;<br>VP3 E59K | 400 |
| PAS32 | Asia1/JS/05 | 2/6 | VP1 P206S | 400 |
| | | 1/6 | VP1 P206T | 400 |
| | | 2/6 | VP1 Q209H | 400 |
| | | 1/6 | VP1 Q209K | 400 |

#The neutralization concentration was determined as the minimum antibody concentration that protected cells from CPE.

$Frequencies of the mutants represent the ratio of escape mutants carrying the indicated substitution(s) to the total number of mutants isolated.

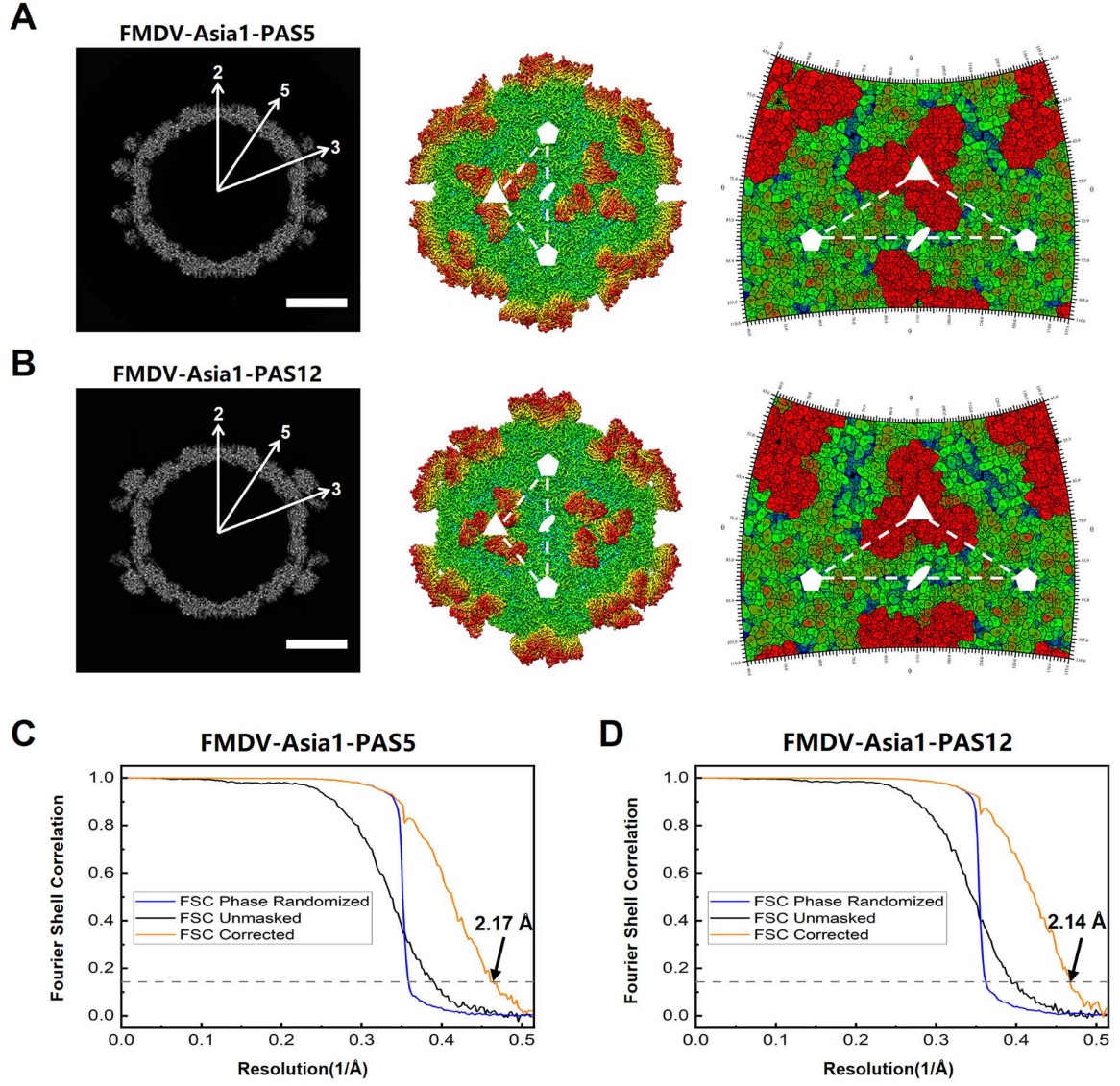

**Fig 4. Cryo-EM analysis of the FMDV-Asia1-PAS5 and FMDV-Asia1-PAS12 complexes. (A-B)** Central cross sections, rendered images, and footprints of the FMDV-Asia1-PAS5 complex (A) or FMDV-Asia1-PAS12 complex **(B)**. The central cross sections obtained from the cryo-EM maps of both complexes display the icosahedral 2-, 3-, and 5-fold axes (Scale bar, 10 nm). In the rendered images, depth cueing with color is used to indicate the radius (< 100 Å, blue; 120-160 Å, from cyan to yellow; > 180 Å, red). The icosahedral 5- and 3-fold axes are represented by pentagons and triangles, respectively. Footprints of PAS5 **(A)** and PAS12 **(B)** on the FMDV serotype Asia1 surface show the 2D projections of the viral surface produced using RIVEM [26]. **(C-D)** The Fourier shell correlation (FSC) curves of the FMDV-Asia1-PAS5 **(C)** and FMDV-Asia1-PAS12 **(D)** complexes, indicating the map resolutions at the 0.143 FSC cutoff.

and $_{VP2}$Y80 form hydrogen bonds with the side chains of $_{VL}$S28 and $_{VL}$S30 in LCDR1, which contains the typical "SXS/R" motif found in all 11 pnAbs recognizing the VP2 B-C loop of FMDV serotype Asia1. Sequence alignment of LCDR1 regions within the whole Asia1-binding BCR repertoire revealed that the "SXS/R" motif was present in more than half of antibodies (54.19%); this comprised primarily the "SXS" motif (49.68%) and a small proportion of the "SXR" motif (4.51%). The "SXS" motif was mainly distributed in LCDR1 regions with lengths of 6 and 9 amino acids, whereas the "SXR" motif was restricted to LCDR1 regions of length 6 (Fig F panels A-C in S1 Text). Given that a length of 6 was observed in

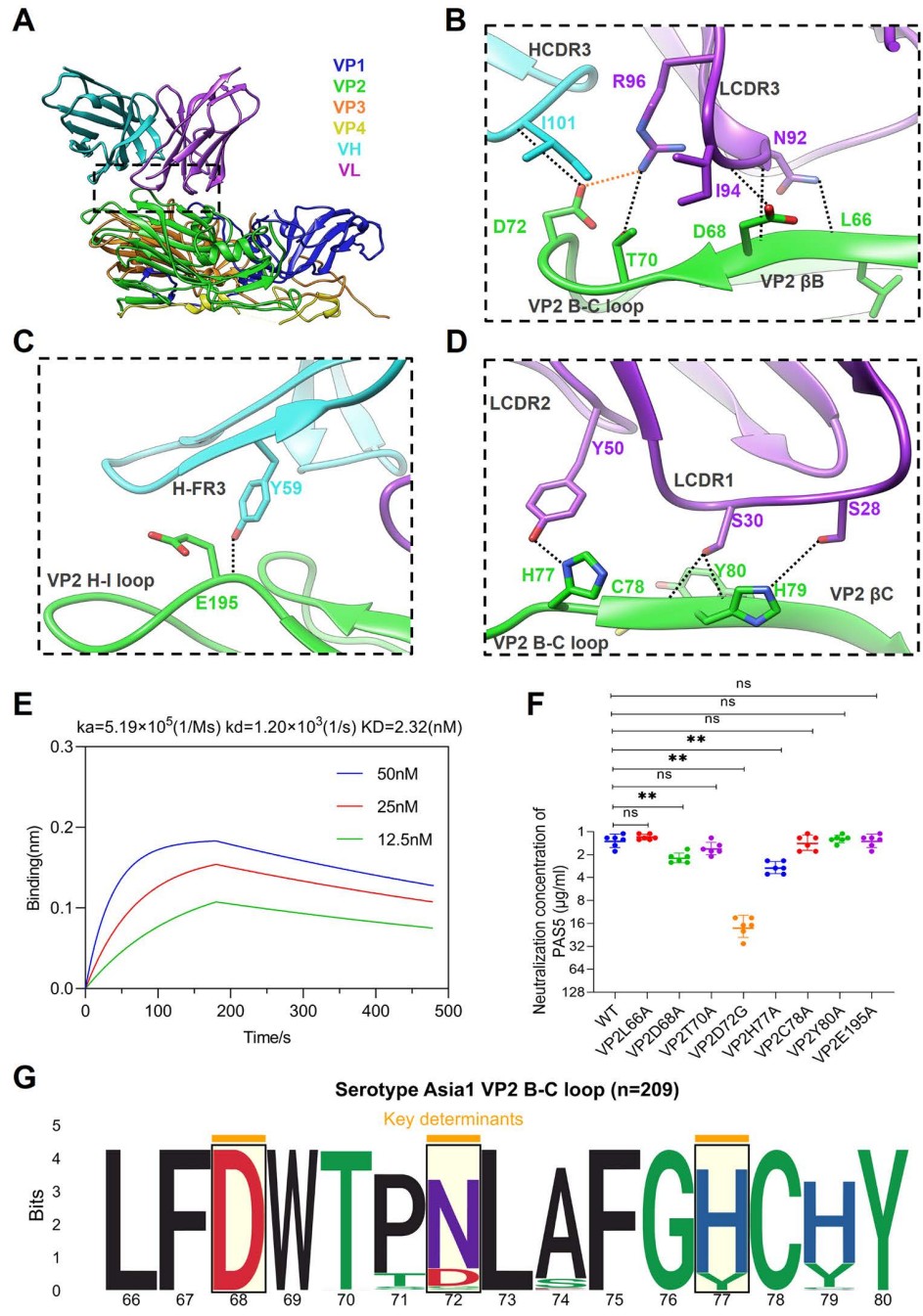

**Fig 5. Structure of the FMDV-Asia1-PAS5 complex and identification of key determinants on VP2 of FMDV serotype Asia1. (A)** Cartoon representation of the interaction interface between the PAS5-scFv and the viral capsid of the Asia1/JS/05 strain within a single protomer. The heavy and light chains of PAS5 are colored cyan and purple, respectively. The viral capsid proteins VP1 to VP4 are colored blue, green, orange, and yellow, respectively. **(B-D)** Expanded views of the PAS5-FMDV interface highlighting the VP2 βB strand, B-C loop **(B)**, H-I loop **(C)**, and βC strand **(D)**. Presumptive hydrogen bonds and salt bridges are marked by black and orange dashed lines, respectively. **(E)** Binding affinity of PAS5 to the inactivated 146S antigen (FMDV Asia1/JS/05 strain) determined by biolayer interferometry (BLI). **(F)** Neutralization potency of PAS5 against the wild-type (WT, Asia1/JS/05 strain) and its single-substitution mutants, evaluated by microneutralization assay. Neutralization concentration represented the minimum antibody concentration required to completely prevent CPE. The experiment was performed in triplicate. Statistical analysis was conducted by One-Way ANOVA followed by Dunnett's multiple comparison test with a 95% confidence interval using GraphPad Prism 8.0. *, **, *** indicate significant differences from WT at $P < 0.05$, $P < 0.01$, $P < 0.001$, respectively. ns indicates no significant difference. **(G)** Sequence conservation analysis of the VP2 B-C loop (residues 66-80) across 209 available FMDV Asia1 sequences.

LCDR1 regions of all 11 pnAbs, the "SXS/R" motif within this length group, encoded by the IGKV1 family (Fig F panels D-E in S1 Text), could serve as a genetic marker for porcine antibodies recognizing the VP2 B-C loop of FMDV serotype Asia1. Generally, the predominant "SXS/R" motif in the BCR repertoire further supported the immunodominance of the VP2 B-C loop on FMDV serotype Asia1.

To determine the crucial determinants on the immunodominant antigenic structure of FMDV serotype Asia1, we introduced amino acid substitutions at capsid residues forming hydrogen bonds or salt bridges at the interface of the FMDV-Asia1-PAS5 complex. A total of eight single-substitution mutants were successfully rescued (Fig G panel A in S1 Text) and assessed for neutralization potency with PAS5. As shown in Fig 5F, substitutions at positions 68, 72, and 77 on VP2 significantly reduced antibody neutralization capacity; most notably, the D72G substitution resulted in a ~16-fold reduction in VN titer. These results indicated that residues D68, D72 and H77 were key determinants within the immunodominant antigenic structure of FMDV serotype Asia1 recognized by PAS5. Furthermore, the sequence analysis of 209 available VP2 sequences of FMDV serotype Asia1 revealed that the conservation rates of residues D68, D72, and H77 were 100%, 15.79%, and 85.65%, respectively (Fig 5G). Notably, the residue at position 72 on VP2 was highly polymorphic, exhibiting multiple amino acid variants: asparagine (N, 80.38%), aspartic acid (D, 15.79%), glycine (G, 1.91%), serine (S, 1.44%), and threonine (T, 0.48%).

**A novel antigenic structure spanning VP2 and VP3 revealed by the cryo-EM structure of the FMDV-Asia1-PAS12 complex**

To provide details of the novel antigenic structure across VP2 and VP3, we resolved the cryo-EM complex structure of PAS12 scFv with Asia1/JS/05 (FMDV-Asia1-PAS12). The cryo-EM reconstruction showed that 60 copies of PAS12-scFv bound to the FMDV-Asia1 capsid, clustering around the icosahedral threefold axis on viral particle (Fig 4B). The final resolution of the cryo-EM reconstruction was estimated at 2.14 Å using the FSC 0.143 cutoff (Fig 4D), which allowed identification of interacting residues at virus-antibody interface at the atomic level.

Structural analysis of the FMDV-Asia1-PAS12 complex (Fig 6A–6D), revealed that PAS12 engages both VP2 and VP3 within a single protomer. The interaction involves the VP2 B-C and H-I loops, along with the VP3 βB strand and B-B knob, and forms interfaces with eight hydrogen bonds (Tables C and D in S1 Text). Binding kinetics analysis indicated that PAS12 exhibits comparable nanomolar affinity for the serotype Asia1 146S antigen with PAS5 (Fig 6E). Notably, the VP2 B-C loop is also present in the interface of the FMDV-Asia1-PAS5 complex. The antibody-interacting residues in the VP2 B-C loop ($_{VP2}$L73, $_{VP2}$S74), H-I loop ($_{VP2}$P186, $_{VP2}$V189), as well as in the VP3 B-B knob ($_{VP3}$E59) and the βB strand ($_{VP3}$K64), interact with residues in both the heavy and light chains, including HCDR3 ($_{VH}$V102, $_{VH}$Y104), HCDR1 ($_{VH}$Y33), HCDR2 ($_{VH}$S52), LCDR1 ($_{VL}$M31, Y33), LCDR3 ($_{VL}$Y92), and H-FR3 ($_{VH}$Y59). Residues $_{VP2}$L73 and $_{VP2}$S74 form hydrogen bonds with $_{VH}$V102. Residues $_{VP2}$P186 and $_{VP2}$V189 form hydrogen bonds with the side chains of $_{VH}$Y104 and $_{VL}$M31, respectively. Residue $_{VP3}$E59 forms hydrogen bonds with the side chains of $_{VH}$Y33, $_{VH}$S52, $_{VH}$Y59. The side chain of $_{VP3}$K64 forms a hydrogen bond with the side chain of $_{VL}$Y92.

To further validate the crucial determinants of the epitope recognized by PAS12 on FMDV serotype Asia1, we introduced amino acid substitutions at FMDV capsid residues involved in forming hydrogen bonds at the interface of the FMDV-Asia1-PAS12 complex. Four single-substitution mutants were successfully rescued and assessed for their neutralization potency with PAS12 (Fig G panel B in S1 Text). As shown in Fig 6F, substitutions at position 73 of VP2 and position 59 of VP3 markedly reduced antibody neutralization. Particularly, the substitution E59K on VP3 resulted in a significant reduction in VN titers (~60-fold). These results indicated that residues $_{VP2}$L73 and $_{VP3}$E59 serve as key determinants of the novel antigenic structure recognized by PAS12 on FMDV serotype Asia1. Furthermore, the conservation rates of residues $_{VP2}$L73 and $_{VP3}$E59 were 100% and 97.13%, respectively (Fig 6G), indicating that the antigenic structure spanning VP2 and VP3 is highly conserved in FMDV serotype Asia1. Additionally, structural comparisons of FMDV-integrin and FMDV-antibody complexes revealed evident steric hindrance between PAS5 or PAS12 and the integrin receptor (αvβ6). This

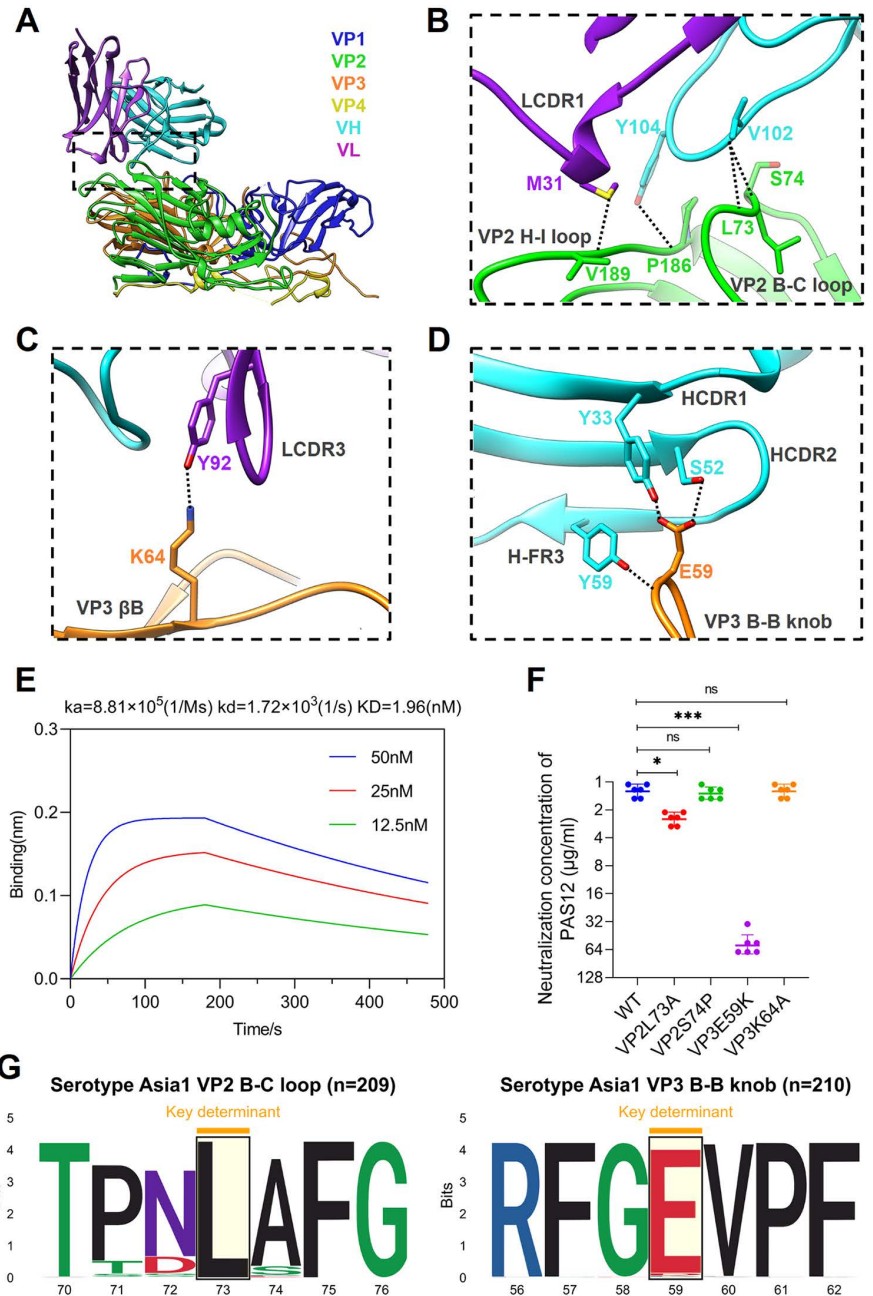

**Fig 6. Structure of the FMDV-Asia1-PAS12 complex and key determinants on VP2 and VP3 of FMDV serotype Asia1. (A)** Cartoon representation of the interaction interface between the PAS12-scFv and the viral capsid of the Asia1/JS/05 strain within a single protomer. The heavy and light chains of PAS12 are colored cyan and purple, respectively. The viral capsid proteins VP1 to VP4 are colored blue, green, orange, and yellow, respectively. **(B-D)** Expanded views of the PAS12-FMDV interface highlighting the VP2 B-C and H-I loops **(B)**, the VP3 βB strand **(C)** and VP3 B-B knob **(D)**. Presumptive hydrogen bonds and salt bridges are marked by black and orange dashed lines, respectively. **(E)** Binding affinity of PAS12 to the inactivated 146S antigen (FMDV Asia1/JS/05 strain) determined by BLI. **(F)** Neutralization potency of PAS12 against the wild-type (WT, Asia1/JS/05 strain) and its single-substitution mutants, evaluated using a microneutralization assay. The neutralization concentration represented the minimum antibody concentration required to completely prevent CPE. The experiments were independently conducted in triplicate. Statistical analysis was performed using One-Way ANOVA followed by Dunnett's multiple-comparisons test with a 95% confidence interval in GraphPad Prism 8.0. *, **, and *** indicate significant difference compared with WT at P < 0.05, P < 0.01, P < 0.001, respectively. ns indicates no significant difference. **(G)** Sequence conservation analysis of the VP2 B-C loop (residues 70-76) and VP3 B-B knob (residues 56-62) among available FMDV Asia1 strains (VP2 sequences = 209; VP3 sequences = 210).

suggested that FMDV neutralization by the PAS5-like and PAS12-like antibodies is mediated by blocking the virus-receptor interaction (Fig H in S1 Text).

**Serological responses reveal a balanced recognition of antigenic sites on VP1, VP2, and VP3 of FMDV serotype Asia1 in pigs following primary and booster vaccination**

To date, four distinct antigenic sites have been mapped on FMDV serotype Asia1 using a panel of 24 pnAbs (Fig 7A and 7B). First, the VP1 G-H loop (containing the core "RGDL" motif) was recognized by pOA20-like antibodies, representing a group of cross-neutralizing pnAbs described previously [17]. Second, the VP1 C-terminus (involving critical residues P206 and Q209) was uniquely targeted by PAS32. Third, an antigenic site on VP2 consisting of the βB strand, B-C loop, βC strand, and H-I loop (with key residues D68, N72, and H77) was recognized by PAS5-like antibodies. Fourth, a novel intra-protomer antigenic site spanning the VP2 B-C/H-I loops (key residue L73) and the VP3 βB strand/B-B knob (key residue E59) was targeted by PAS12-like antibodies. The antigenic footprints of PAS5 partially overlapped with those of PAS12; consequently, steric clashes between PAS5 and PAS12 were observed upon superposing the FMDV-Asia1-PAS12 and FMDV-Asia1-PAS5 complexes (Fig 7C). The conservation of the identified antigenic sites was evaluated across FMDV serotypes Asia1, O, and A, based on available sequences from GenBank. As shown in Fig I in S1 Text, residue D68 on VP2, as well as residues P206 and Q209 on the VP1 C-terminus, were highly conserved. In contrast, positions 72, 73, and 77 in the VP2 B-C loop and position 59 in the VP3 B-B knob exhibited significant serotype-specific variation. These residues indicated antigenic differences among the three serotypes.

To further determine the immunodominance of these antigenic sites on FMDV serotype Asia1, ten pigs were immunized with an inactivated FMDV serotype Asia1 vaccine formulated with whole 146S antigen and ISA201 adjuvant (Fig 7D). Vaccination with the serotype Asia1 vaccine induced a neutralizing antibody response by day 7, and the neutralization titers increased significantly after the booster vaccination (Fig 7E). The relative antibody responses against each antigenic site in pigs were quantified by competitive ELISA (cELISA) using the pnAbs pOA-20, PAS5, PAS12, and PAS32, which respectively target four distinct antigenic regions on FMDV serotype Asia1 (Fig 7A).

Following primary vaccination, the antibody response to the VP1 G-H loop was comparable to that against antigenic site 2 on VP2 and the novel intra-protomer site spanning VP2 and VP3; notably, these responses were significantly higher than that against the VP1 C-terminus (Fig 7F). Following the booster vaccination, antibody responses to all antigenic sites of FMDV serotype Asia1 were increased, resulting in a balanced antibody response profile against the three antigenic sites on VP1, VP2, and VP3 (Fig 7G). These data demonstrated a balanced antibody response profile, which implies a potentially balanced immunodominance among the predominant antigenic sites on VP1, VP2, and VP3 of FMDV serotype Asia1. Regarding the serum from pig #2219 (the source of the Asia1-binding BCR repertoire), the competitive antibody titers against pOA-20, PAS32, PAS5, and PAS12 were 1: 128, 1: 45, 1: 512, and 1: 512, respectively. The relatively low antibody titers targeting VP1 in this pig likely explain the preponderance of isolated antibodies against VP2 and VP3 rather than VP1, reflecting the individual variability in the BCR repertoire constructed from a single animal.

To further compare the antibody response profiles among FMDV serotypes, sera from pigs vaccinated with serotypes A and O were analyzed using cELISA. Both serotype A- and O-vaccinated porcine sera showed predominant antibody recognition targeting the VP1 G-H loop. A secondary response was observed to antigenic site 2, whereas antibody titers to epitopes on VP3 were the lowest, indicating a clear hierarchy in epitope immunodominance across these serotypes (Fig J and Table E in S1 Text).

Collectively, we mapped four distinct antigenic sites on FMDV serotype Asia1 using pnAbs, which characterized the antigenic landscape of the eradicated virus. In contrast to serotypes O and A, FMDV serotype Asia1 elicited a balanced antibody response, suggesting a more evenly distributed immunodominance of antigenic sites across its structural proteins. This distinct antigenic profile was associated with a broader antibody response and might contribute to the effective immunogenicity of the vaccine antigen.

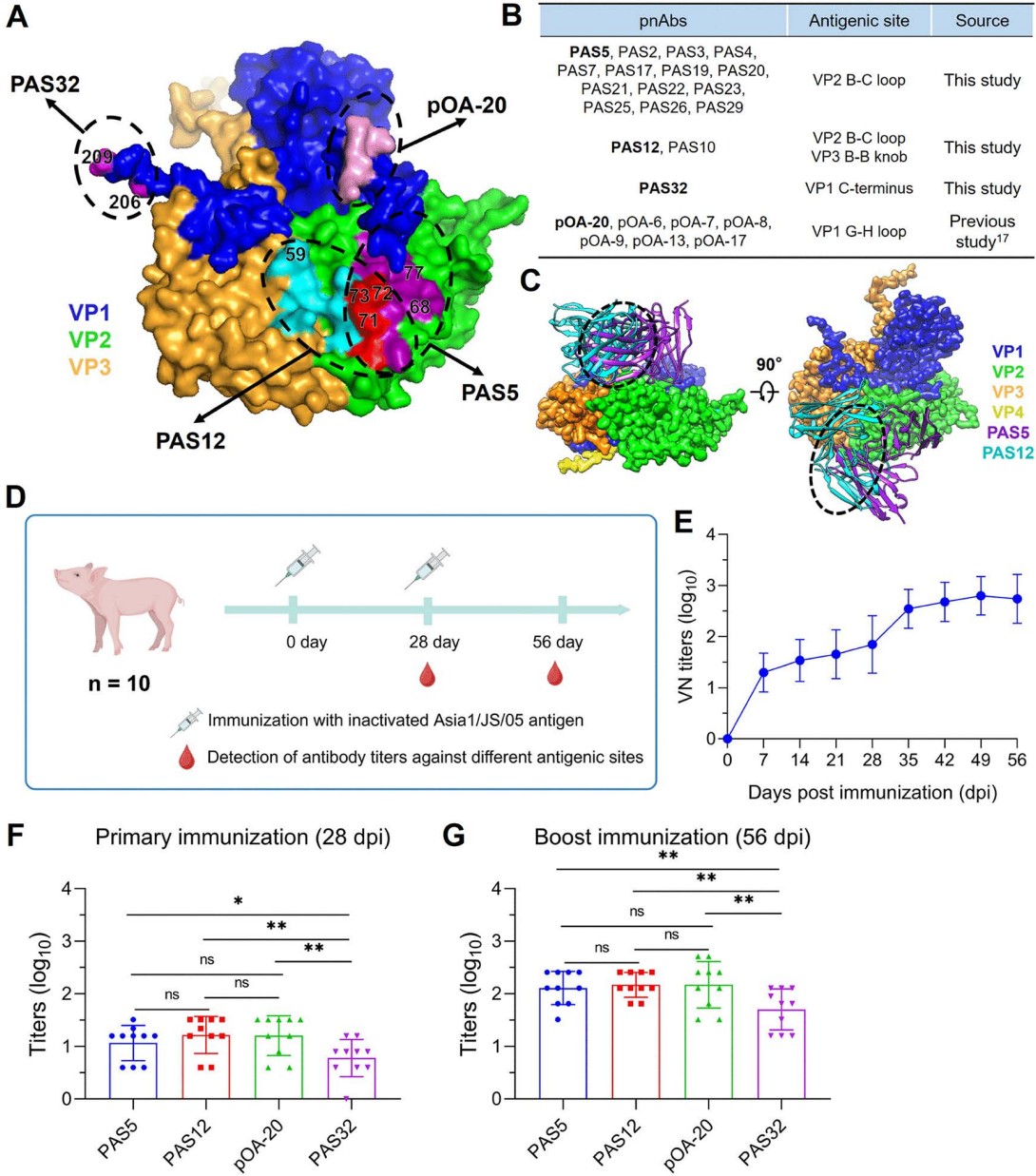

**Fig 7. Identification of immunodominant antigenic sites on FMDV serotype Asia1. (A)** Footprints of four distinct antigenic sites on FMDV serotype Asia1 identified by pnAbs. The capsid proteins VP1, VP2, and VP3 are colored blue, green, and orange, respectively. The residues of the "RGDL" motif in the VP1 G-H loop are marked in pink, and the key determinants 206 and 209 in the VP1 C-terminus are shown in magenta. The common residues recognized by PAS5 and PAS12 are shown in red. The rest of PAS5-interacting residues on VP2 are marked in purple, including three key residues 68, 72 and 77. The rest of PAS12-interacting residues are marked in cyan, including residues 73 on VP2 and 59 on VP3. **(B)** Characterization of pnAbs targeting four distinct antigenic sites on FMDV serotype Asia1. **(C)** Structural modeling of steric hindrance between PAS5 and PAS12 bound to FMDV Asia1/JS/05. VP1, VP2, VP3, and VP4 are colored blue, green, orange, and yellow, respectively. PAS5 and PAS12 are colored in purple and cyan, respectively. Black dashed circles indicate significant clashes between PAS5 and PAS12. **(D)** Schematic diagram of the immunization and sampling schedule for 10 pigs immunized with the inactivated Asia1/JS/05 vaccine. Created in BioRender. Chen, Y. (2026) https://BioRender.com/vathqjk. **(E)** Dynamics of neutralizing antibody titers against the Asia1/JS/05 strain in serum samples collected at different time points post-vaccination. **(F-G)** Competitive ELISA detection of antibody titers against different antigenic sites in serum samples collected at 28 dpi **(F)** and 56 dpi **(G)** from the 10 pigs. Statistical analysis was performed using One-Way ANOVA followed by Dunnett's multiple-comparison test with a 95% confidence interval in GraphPad Prism 8.0. ns indicates no significant difference. *, **, *** indicate significant difference at $P < 0.05$, $P < 0.01$, $P < 0.001$, respectively.

## Discussion

In summary, we established a natural host BCR repertoire binding to FMDV Asia1/JS/05 strain and identified a panel of pnAbs. A total of four antigenic sites on FMDV serotype Asia1 were mapped by 17 newly identified pnAbs from the BCR repertoire and 7 cross-neutralizing pnAbs from our previous report [17]. The antibody response targeting the VP1 G-H loop, antigenic site 2, and a novel site spanning sites 2 and 4 was evaluated using sera from 10 pigs vaccinated with the Asia1/JS/05 inactivated vaccine, revealing a balanced antibody response profile that implies a potentially evenly distributed immunodominance among antigenic sites on VP1, VP2, and VP3 of serotype Asia1.

Humoral immunity is often characterized by dominant or recessive responses to different epitopes on the same antigen [27–29]. The shape and valency of immunogens influence B cell immunodominance patterns, the selection process within germinal centers, and the expansion of memory B cell populations [30]. Extensive research on the immunodominance of antigenic sites on FMDV serotypes O and A has revealed the focused antigen distribution features. The G-H loop of VP1 has been identified as an immunodominant epitope on FMDV using peptides [31–34]. Nevertheless, various peptide vaccines based on antigenic site 1 have shown limited efficacy [35]. Additionally, the immunodominance of site 2 and/or site 1 has been demonstrated in certain strains of FMDV serotype O [36–38]. In this study, sera from pigs vaccinated with FMDV serotypes A and O showed that the VP1 G-H loop (Site 1) was the primary immunodominant region. Essentially, the presence of excessive antibodies targeting the immunodominant sites may sterically hinder the surrounding non-immunodominant epitopes, blocking the animal immune system from producing sufficient antibody responses targeting non-immunodominant sites, and possibly limit the efficacy of vaccination [39]. In contrast, FMDV serotype Asia1 elicited a more balanced antibody response targeting antigenic sites on VP1, VP2, and VP3. This suggests a potentially even distribution of immunodominance across viral surface, which may facilitate the utilization of a broader range of germline gene segments for recognizing diverse epitopes. However, we acknowledge that the cELISA format used in this study may affect epitope accessibility, due to potential conformational changes or steric hindrance introduced by the capture antibody (mAb E32). Therefore, definitive confirmation requires further validation using solution-phase assays or structural techniques.

Traditionally defined antigenic site 2 forms conformational epitopes on FMDV serotypes A and O [40–42]. Notably, we found PAS5-like antibodies could bind with denatured 146S antigen and VP2 protein of Asia1/JS/05 strain, but it did not react with further truncated VP2 protein. These results revealed site 2 of FMDV serotype Asia1 forms incomplete linear antigenic structure. The interface residues within the conformational antigenic site 2 exclusively locate to the B-C/E-F/H-I loops of VP2 for serotype O, whereas in serotype A, they are primarily focused on the B-C/H-I loops of VP2 and an additional key residue 61 on the VP3 B-B knob [13–15]. However, pnAb PAS5 revealed that the key residues of incomplete linear antigenic site 2 were primarily concentrated in the βB strand, B-C loop, and βC strand. This indicates a structural difference in antigenic site 2 among FMDV serotype A, O, and Asia1. The incomplete linear epitopes of antigenic site 2 potentially represent another specific structure property on FMDV serotype Asia1.

The porcine heavy chain variable region locus contains 21 $V_H$ genes as reported in IMGT GENE-DB (Version 3.1.42) [43]. The BCR repertoire binding to FMDV serotype Asia1 utilizes a set of 15 $V_H$ genes, with only 5 comprising approximately 86.95% of the repertoire, which is consistent with FMDV serotypes A and O [17]. Regarding the light chain, all three repertoires utilize a consistent set of nine kappa genes belonging to the IGKV1 and IGKV2 families, as well as eight lambda genes belonging to the IGLV3 and IGLV8 families. However, the Asia1-binding BCRs predominantly utilize the κ chain (63.0%), while BCRs elicited by sequential immunization with serotypes A and O exhibit a higher usage of the lambda chain (75.5%). Unlike humans and mice, porcine light chain rearrangement precedes heavy chain rearrangement [44]. Specifically, initial kappa rearrangements are rapidly replaced by lambda rearrangements before recombination of heavy chain genes. This rearrangement mechanism may cause selective pressure for higher diversification of lambda locus because the availability of more genes allows for a higher degree of editing and repertoire diversification [22]. Hence, the multiple sequential immunizations with FMDV serotypes A and O probably induce a higher selective pressure

on the lambda locus compared to only two immunizations with serotype Asia1. Given that these results are based on a single animal, further validation in a larger cohort is required to confirm their generalizability.

Conventionally, the role of the light chain in the structure of antibodies appears to be to stabilize the binding site, while the CDR3 region of the heavy chain contributes significantly to antibody diversity and provides energy for antigen binding [45,46]. Previous cryo-EM structures of FMDV-antibody complexes revealed that heavy chains were mainly involved in forming hydrogen bonds and salt bridges with antigenic epitopes [13–15]. Interestingly, we found that the light chain CDR regions of porcine antibody PAS5 primarily forms both hydrogen bonds and a salt bridge with FMDV Asia1/JS/05. This finding confirms the important role of the light chain in neutralizing activity and diversity of porcine antibodies. Notably, residues C78, H79 and Y80 in VP2 were specifically targeted by the "SXS" motif in LCDR1 of PAS5 through hydrogen bonds. We found a common "SXS/R" motif in the 6-residue LCDR1 region of PAS5-like antibodies originating from the IGKV1 gene family. This is the first report describing a hotspot motif in the light chain of porcine antibodies for viral recognition. In contrast, hotspot motifs in the heavy chain CDR regions that specifically bind to or neutralize the virus has been identified in multiple other viruses. For instance, a $CX_4C$ motif in the HCDR3 region of multiple Nabs was found to be critical for the neutralization activity against Hepatitis C virus (HCV) [47]. For SARS-CoV2, IGHV3–53/3–66 antibodies carry germline-encoded features, namely an NY motif in HCDR1 and an SGGS motif in HCDR2, which are critical for RBD binding [48–50]. Several of bnAbs against SARS-CoV2 and SARS-CoV utilize a common motif in antibody HCDR3 region, the 'YYDRSGY' motif, which originates from IGHD3–22 gene [51].

Despite these insights, there are several inherent limitations in our immunogenetic characterization of the porcine B-cell repertoire against FMDV serotype Asia1. First, BCR sequencing and antibody isolation were derived from a single pig, which may not fully capture the interindividual variability inherent to outbred pigs, given the extensive genetic diversity of porcine immunoglobulin loci. Second, the overall repertoire diversity may be limited by both the sequencing depth and the stringent clonotype definition criteria, potentially excluding clonotypes with minor somatic variations but convergent antigen specificity. Third, only a subset of dominant clonotypes from the single pig, in which the response to antigenic site 2 was predominant, was recombinantly expressed and functionally characterized. Consequently, this approach may not fully capture the complete antibody landscape and could underrepresent antibodies targeting other potentially dominant epitopes, such as the VP1 G-H loop. As such, this study should be regarded as a preliminary exploration, and future studies involving multiple animals and broader repertoire coverage are warranted to gain a more complete understanding of porcine humoral responses to FMDV Asia1. Nonetheless, our findings provide valuable insights into the structural and immunogenetic determinants of neutralizing antibodies, offering a foundation for rational vaccine design and viral eradication strategies.

# Materials and methods

## Ethics statement

All the animal experiments conducted in this study were approved by the Review Board of Lanzhou Veterinary Research Institute, Chinese Academy of Agricultural Sciences (Permit No. LVRIAEC2022–016), and were performed according to the Animal Ethics Procedures and Guidelines of the People's Republic of China.

## Virus production, purification and biotinylation of 146S antigen

FMDV Asia1/JS/05 antigens were produced and harvested in baby hamster kidney cells (BHK-21). Viral supernatant was inactivated by BEI (binary ethylenimine) for 28 h at 30°C. For further purification, the 146S antigen were first concentrated by 8% (w/v) PEG 6,000 at 4°C overnight and centrifugated at 3,500 × g for 1 h, followed by resuspension in PBS (pH = 7.8). The resuspension was pelleted subsequently through a cushion of 30% (wt/vol) sucrose in PBS (pH = 7.8) by centrifugation at 35,000 × g for 1.5 h at 4°C. The pellet was resuspended in 500 µl of PBS (pH = 7.8) and fractionated by

centrifugation at 35,000 × g for 4 h at 4°C. The fractions were analyzed by negative stain electron microscopy, and the fraction containing 146S particles was transferred to a 100 kDa MWCO centrifugal filter for buffer exchange with PBS (pH = 7.8) to remove the sucrose. The final 146S particles were quantified by absorbance at 260 nm (where an optical density of 7.6 = 1 mg/ml) and immediately used for subsequent experiments. Highly purified FMDV (Asia1/JS/05) 146S antigen were incubated with NHS-LC-Biotin reagent (Thermo Fisher Scientific, USA) in accordance with the manufacturer's instructions.

### Porcine vaccination

A three-month-old healthy pig, #2219, was raised in a clean animal facility and immunized intramuscularly inactivated FMDV serotype Asia1 (Asia1/JS/05 strain) vaccine twice with one dose (20 µg 146S antigen formulated with ISA 201 adjuvant) at four-week interval. On day 5 after the secondary vaccination, about 200 ml of the EDTA-anticoagulated peripheral blood sample was collected. Subsequently, porcine PBMCs were isolated using HISTOPAQUE 1.077 (Sigma-Aldrich, USA) for further experiments. Additionally, a group of ten healthy pigs, which are negative for FMDV 3ABC antibody, were selected for evaluating the immunodominance of antigenic structures on FMDV serotype Asia1. These pigs were twice immunized with the inactivated FMDV vaccine (Asia1/JS/05 strain) as above immune procedure.

### Purification of FMDV serotype Asia1-binding B cell from porcine PBMCs

Antigen-binding B cells of FMDV Asia1/JS/05 strain were isolated and purified from porcine PBMCs using negative magnetic separation and FACS following previous description [17]. Briefly, approximately $8.0 \times 10^8$ porcine PBMCs were initially suspended in 5 ml RPMI 1640 medium and incubated with 25 µl each of mouse anti-pig CD3 antibody, mouse anti-pig CD14 antibody, mouse anti-pig CD335 antibody and mouse anti-pig IgM antibody (Bio-Rad) on ice for 30 min. After twice washes, above PBMCs were resuspended and incubated in 2 ml goat anti-mouse IgG microbeads on ice for 30 min. After twice washes, those PBMCs were further loaded into an LD column for negative separation. Subsequently, the effluent cells were collected for further staining. The biotinylated 146S antigen (Asia1/JS/05 strain) were employed as bait to capture antigen-binding B cells, followed by incubation with anti-biotin-APC antibody. Finally, the stained cells were promptly loaded on flow cytometry (BD FACSAria II, USA) to sort sufficient Asia/JS/05-binding cells.

### Construction and sequencing of porcine single B cell library

After purification by FACS, Asia1/JS/05-binding single cells were initially resuspended in PBS buffer (pH = 7.4). Single-cell capture and library construction were conducted using the 10 × Genomics Chromium Next GEM Single Cell 5' Kit v2 (PN-1000263) with a target loading of about 10,000 cells per reaction according to the manufacturer's protocols. Briefly, the cell suspension, barcoded gel beads, and partitioning oil were loaded onto the Chromium Chip K to generate single-cell Gel Beads-in-Emulsion (GEMs). Captured cells were lysed, and the transcripts were barcoded via reverse transcription inside individual GEMs. The cDNA along with cell barcodes were then PCR-amplified. The scRNA-seq library was performed by using the 5' Library Kits (PN-1000190), and the scBCR-seq library was constructed by using the Human B Cell V(D)J Enrichment Kits (PN-1000252), with custom porcine BCR primers panel replacing the V(D)J fragments primers as previous description [17]. The libraries were sequenced on Illumina NovaSeq 6000 platform to produce 2 × 150-bp paired-end reads.

### Porcine B cell receptor repertoire analysis

BCR sequences were assembled and quantified according to the Cell Ranger (v.4.0.0) vdj protocol against porcine V(D)J reference created with the command-line tool mkvdjref utilizing IMGT/V-QUEST reference *(Sus scrofa)* as input. Assembled contigs labeled as low-confidence, non-productive, or with fewer than 2 UMIs were discarded. The resulting

filtered_contig_annotations.csv and filtered_contig.fasta files were reanalyzed using R (v4.3.1) to identify paired clonotypes. These clonotypes were defined as B cell clone containing exactly paired heavy and light chains with identical HCDR3 + LCDR3 amino acid sequences. The frequency of clonotype was determined by counting the number of distinct cell barcodes for each unique HCDR3 + LCDR3. Clonotypes supported by only one cell were classified as unexpanded, while those clones supported by two or more cells were classified as expanded.

The germline gene lineage and SHM of porcine antibodies were reanalyzed using the Immcantation program. We aligned the paired BCR contigs to IMGT reference genes using HighV-Quest, and the output was parsed with Change-O [52]. The SHM was then calculated using the SHazaM tool [53].

## Expression and purification of porcine mAbs

The selected paired $V_L$ and $V_H$ genes were codon-optimized with *Cricetulus griseus* and separately inserted into constructed porcine heavy chain (CH-pcDNA3.4) and light chain (Cκ-pcDNA3.4 or Cλ-pcDNA3.4) expression vectors to synthesize the full-length antibody-expressing plasmids, as described in previous report [54]. The recombinant scFv was designed by linking antibody $V_H$ and $V_L$ fragments with a $(GGGGS)_3$ spacer and adding a hexahistidine tag (HHHHHH) at the C-terminus. The modified scFv genes were then cloned into expression vector pcDNA3.4, respectively.

The antibody-expressing plasmids were respectively transfected into the suspended 293F cells (Invitrogen, USA) with a heavy-to-light chain ratio of 2:3, followed by continued cultivation for a week. The expressed mAbs in supernatants were first purified through Ni-chelating affinity chromatography. The scFvs were further purified by size exclusion chromatography with a Superdex 200 increase 10/300 column on an AKTA plus protein purification system (GE Life Sciences). The concentration of purified mAbs was determined by quantifying the absorbance values at 280 nm (A280).

## ELISA

The ELISA plates were coated with inactivated 146S antigen of Asia1/JS/05, and then probed with different concentrations of 0–20 µg/ml of the tested pnAbs, followed by probing with HRP-conjugated goat anti-porcine IgG. Color was developed by adding 50 µl of TMB substrate (Pierce, Life Technology) for 10 min at room temperature. The process was stopped by adding equal volumes of 1M $H_2SO_4$. Optical density at 450 nm ($OD_{450}$) was measured on a microplate reader (BioRad). The experiments were independently conducted in triplicate.

## Virus neutralization test

The porcine mAbs or serum samples were titrated for viral neutralizing activity against FMDV serotype Asia1 (Asia1/JS/05 strain) or the rescued virus by using a micro-neutralization assay as previously described [55]. Briefly, mAbs or serum samples were 2-fold serially diluted in 96-well cell culture plates in a total volume of 50 µl, and then 100 tissue culture infectious dose ($TCID_{50}$) of FMDV in 50 µl of culture media was added to each well. After incubation for 1 h at 37°C, ~$5 \times 10^4$ BHK-21 cells in 100 µl media were added to each well as indicators of residual infectivity. Normal cell wells, 0.1, 1, 10, and 100 $TCID_{50}$ virus control wells in duplicate were used in each plate. The plates were incubated at 37°C under 5% $CO_2$ conditions for 48~72 h before observing cytopathic effect (CPE). The experimental results were acceptable when complete CPE and no CPE appeared separately in 100 $TCID_{50}$ and 0.1 $TCID_{50}$ virus control wells. The endpoint titers were calculated as the reciprocal of the last serum dilution to neutralize 100 $TCID_{50}$ FMDV in 50% of the wells. Neutralizing activity is expressed as the VN titer, which was calculated as the initial antibody concentration divided by the endpoint titer.

## Selection of neutralization-escape mutants using porcine mAbs

Neutralization escape mutants were generated by consecutive passages of FMDV in BHK-21 cells under the selective pressure of pnAbs, following a previously reported protocol with minor modifications [17]. The representative FMDV

serotype Asia1 strain Asia1/JS/05, was utilized to select mutants against these pnAbs. Briefly, 10-fold serial dilutions of FMDV in 50 μl were incubated with 50 μl of various concentrations of pnAbs (ranging from 20 to 50 μg/ml) in 96-well microplates. Subsequently, the mixtures were used to infect BHK-21 cells ($10^6$ cells/ml) in a volume of 100 μl and incubated at 37°C for 48~72 h to allow virus propagation. First-passage viruses were harvested from wells seeded with the highest dilution of virus that produced an approximately 80–100% CPE. Further rounds of pressure selection were performed in 24-well plates, in which the passaged virus (200 μl) was incubated with an equal volume of a 2-fold concentration of antibodies in each well containing BHK-21 cells (400 μl). The harvested virus was subjected to several additional rounds of selection until it completely escaped neutralization after the addition of pnAbs at concentrations of at least 400 μg/ml. The P1 region sequence corresponding to the obtained neutralization escape mutants was amplified by one-step reverse transcription-PCR (RT-PCR) as described previously, using the primer pair Pan204+ (ACCTCCAACGGGTGG-TACGC)/NK61 (GACATGTCCTCTTGCATCTG) [56]. Subsequently, the amplified products were verified by sequencing. Mutated amino acids were determined by aligning the entire mutant P1 region with its initial parent virus sequence.

### Cryo-EM samples preparation and data collection

Among the neutralizing antibodies isolated from the Asia1/JS/05-binding B cell repertoire, PAS5 and PAS12 were selected for cryo-EM analysis based on two main criteria: (i) potent neutralizing activity against FMDV Asia1/JS/05 and (ii) recognition of distinct antigen sites according to primary results by selecting neutralization escape mutants.

FMDV Asia1/JS/05 146S was individually incubated with scFv (PAS5-scFv or PAS12-scFv) at a molar ratio of 1:60 in a volume of 10 μl. A 4-μl aliquot of the mixture was applied to a glow-discharged 400 mesh grid (Quantifoil Au R1.2/1.3) supported with a thin layer of Graphene Oxide. Grids were blotted in a Vitrobot mark IV (Thermo Fisher, USA) at 4°C and 100% humidity for 5 s prior to plunging into liquid ethane. Images of complex were acquired at 300 kV with a Titan Krios G3i (Thermo Fisher, USA) and a direct electron detector (K3 Bioquantum, Gatan) at Lanzhou University. Images were recorded at −1.8 to −0.8 μm under focus at a calibrated magnification of ×105 kX, resulting in a pixel size of 0.83 Å. Each image was dose-fractionated into 20 movie frames with a total exposure time of 1.5 s.

### Image processing and three-dimensional reconstruction

For the datasets of FMDV-Asia1-PAS5 and FMDV-Asia1-PAS12, movie stacks were motion corrected and electron-dose weighted using Patch-Based Motion Correction in cryoSPARC [57]. Individual frames from each movie were aligned and averaged to create drift-corrected images. Particles were automatically picked and selected, and contrast transfer function (CTF) parameters were estimated using CTFFIND4 [58]. The following ab-initio, homogeneous, and non-uniform reconstructions were all carried out in cryoSPARC. For all reconstructions, the resolution was assessed using the standard FSC = 0.143 criterion. The data collection and refinement statistics are summarized in Table F in S1 Text.

### Model building and refinement

To construct initial models of FMDV-Asia1-PAS5 and FMDV-Asia1-PAS12, the predicted structure of FMDV-Asia1 by using SWISS-MODEL and the predicted structure of PAS5 or PAS12 using AlphaFold2 were initially docked into density maps with UCSF Chimera [59,60]. Two complex models were refined over multiple rounds with real-space refinement using Phenix in combination with manual adjustment in Coot [61,62]. The quality of all refined models was assessed using Mol-Probity [63].

### Bio-layer interferometry (BLI) assay

The binding affinity of PAS5 or PAS12 to FMDV (serotype Asia1) was determined using BLI assay in the GatorPrime biosensor system (GatorBio) according to manufacturer's instructions. Briefly, biotinylated Asia1/JS/05 146S (10 μg/ml)

was immobilized onto streptavidin-coated biosensors (Gator Bio, 20–5016) until saturation. The antigen-bound biosensors were placed in wells containing a series of diluted PAS5 or PAS12 to allow antigen-antibody association. Subsequently, the biosensors were immersed in dissociation buffer (0.01 M PBS added with 0.1% bovine serum albumin and 0.02% Tween 20). The on-rate (ka) and off-rate (kd) were determined through global fitting of the association and dissociation phases across various pnAbs concentrations. The equilibrium dissociation constant (KD), which measures affinity, was then calculated as the ratio of kd and ka.

### Rescue of site-directed FMDV mutants by reverse genetics

The full-length cDNAs were generated by using an existing Asia1/JSp1 plasmid containing the whole P1 gene of FMDV Asia1/JS/05 [64]. Site-directed mutagenesis was subsequently employed to introduce nucleic acid mutations, resulting in the production of full-length cDNAs with single amino-acid substitutions [65]. All mutant constructs were validated through nucleotide sequencing. The site-directed FMDV mutant viruses were rescued as previously reported [66]. Briefly, *Not* I-linearized mutant plasmids were transfected into BSR/T7 cells using Lipofectamine 2000 following the manufacturer's instructions. The transfected cells were monitored daily for appearance of CPE. At 72 h post-transfection, the cells were harvested and passaged in BHK-21 cells. After 3 rounds of passaging, the mutant virus titers were determined in BHK-21 cells by calculating the 50% $TCID_{50}$, which was subsequently used to perform micro-neutralization assay as described above.

### Competitive ELISA

A capture cELISA was performed to evaluate antibody responses targeting distinct antigenic sites. Briefly, 96-well plates were coated overnight at 4°C with E32, a bovine broadly reactive FMDV monoclonal antibody, at 0.5 µg/ml in carbonate-bicarbonate buffer (pH = 9.6). After three washes with PBST, 146S antigen (1 µg/ml) was added and incubated for 2 h at room temperature. Plates were blocked with PBS containing 5% sucrose and 1% BSA for 1 h at 37°C. To determine the optimal concentration of biotinylated neutralizing antibodies (Biotin-nAbs), Biotin-nAbs were pre-diluted to 5 µg/ml and subjected to eight successive twofold serial dilutions. Plates were incubated with each dilution for 1 h at 37°C, washed, incubated with HRP-conjugated streptavidin (1: 30000) for 30 min, washed again, developed with TMB for 15 min at 37°C, stopped with 2 M $H_2SO_4$, and read at 450 nm. The dilution producing an $OD_{450nm}$ of ~2.0 was selected as the working concentration.

For competition assays, 50 µl of Biotin-nAb at twice this working concentration was mixed with 50 µl of serum (serial dilutions 1: 8–1: 1024) in antigen-coated wells and incubated for 1 h at 37°C. Plates were washed and detection was performed as described above. Each dilution was tested in duplicate. PBS served as the negative control and Biotin-nAb alone ($OD_{450nm} \approx 2.0$) served as the positive control. The inhibition rate was calculated as (average OD of positive control - OD of sample well)/ (average OD of positive control - average OD of negative control). The highest serum dilution yielding >50% inhibition was recorded as the antibody titer.

### Supporting information

**S1 Text.** Fig A. Negative-stain electron microscopy analysis of the particle integrity of purified FMDV 146S antigens before and after biotinylation. Negative-stain electron microscopy analysis of purified Asia1/JS/05 146S particles (A) and the corresponding biotinylated Asia1/JS/05 146S particles(B). Fig B. Reactivity of porcine mAbs with FMDV serotype Asia1 detected by indirect immunofluorescence assay (IFA). BHK-21 cells were infected with the Asia1/JS/05 strain and incubated with porcine mAbs (5 µg/ml), followed by incubation with FITC-conjugated rabbit anti-pig IgG (1: 200 in PBS). Fluorescence signals were observed using an FL Imaging System (Life Technologies, USA). Experiments were independently conducted in triplicate. Fig C. Reactivity of porcine mAbs with FMDV serotype Asia1 determined by indirect enzyme-linked

immunosorbent assay (ELISA). Fig D. Neutralizing potency of pnAbs against FMDV Asia1/JS/05 strain and its mutants was evaluated using a microneutralization assay. The neutralizing concentration represented the minimum antibody concentration required to fully prevent CPE. The experiment was performed in triplicate. Statistical analysis was conducted by One-Way ANOVA followed by Dunnett's multiple comparison test or unpaired T-test with a 95% confidence interval using GraphPad Prism 8.0. *, **, *** indicate significant differences from WT at $P < 0.05$, $P < 0.01$, $P < 0.001$, respectively. ns indicates no significant difference. Fig E. Reactivity of pnAbs PAS5 and PAS12 against denatured 146S antigen of the Asia1/JS/05 strain by Western blotting. The 146S antigen of the Asia1/JS/05 strain was denatured and reduced by heating at 100°C for 5 min in SDS-loading buffer with dithiothreitol (DTT), separated by 12% SDS-PAGE, and transferred to a methanol-activated nitrocellulose membrane. After blocking with 5% non-fat milk in TBST overnight at 4°C, membranes were sequentially incubated with porcine mAbs PAS5 or PAS12 (2 µg/ml) and HRP-conjugated anti-porcine IgG (1: 5000) for 1 h at 37°C. Signals were visualized using an enhanced chemiluminescence solution (Thermo Fisher Scientific, USA) for 1 min and subsequently exposed to X-ray film. Fig F. Characterization of "SXS" and "SXR" motifs in the LCDR1 region of the Asia1/JS/05-binding BCR repertoire. (A-B) Distributions of amino acid (AA) lengths in LCDR1 regions containing "SXS" (A) or "SXR" (B) motifs. (C) AA sequence alignment of the "SXS" and "SXR" motifs in 6-residue LCDR1 regions. Motifs were boxed in black and shaded in light blue ("SXS") and light coral ("SXR"). (D) Usage of light chain V gene segments in 9-residue LCDR1 regions containing the "SXS" motif. (E-F) Usage of light chain V gene segments in 6-residue LCDR1 regions containing the "SXS" motif (E) and the "SXR" motif (F). Fig G. Plaque phenotype of wild-type (Asia1/JS/05) and rescued single-substitution mutants. (A) Plaque formation of the WT (Asia1/JS/05) and rescued single-substitution mutants (VP2 L66A, VP2 D68A, VP2 T70A, VP2 D72G, VP2 H77A, VP2 C78A, VP2 Y80A and VP2 E195A). (B) Plaque formation of the WT (Asia1/JS/05) and rescued single-substitution mutants (VP2 L73A, VP2 S74P, VP3 E59K and VP3 K64A). Fig H. Binding modes of FMDV with integrin receptor and antibody. The FMDV-receptor complex structure was determined by Kotecha A et al. in a previous study. (A-B) Superposition of FMDV-avβ6 integrin complexes with FMDV-Asia1-PAS5 (A) or FMDV-Asia1-PAS12 (B). VP1, VP2, VP3, and VP4 of the protomer are shown in blue, green, orange, and yellow, respectively. The av and β6 chains of integrin and PAS5 or PAS12 are drawn in cartoon representation and colored in magenta, cyan, and purple, respectively. Black dashed circles show significant clashes between the antibody (PAS5 or PAS12) and the integrin receptor. Fig I. Sequence conservation of key antigenic sites among FMDV serotypes Asia1, A, and O. Sequence logos illustrate amino acid conservation patterns at three major antigenic regions across representative strains of serotypes Asia1, A, and O. (A) VP2 B-C loop, (B) VP3 B-B knob, and (C) VP1 C-terminus. The height of each amino acid symbol reflects its relative frequency at that position, indicating sequence conservation or variability among serotypes. Key residues identified in this study are highlighted with red boxes. Fig J. Competitive ELISA analysis of sera from pigs immunized with FMDV serotypes A and O. Competitive ELISA was performed to evaluate the serum antibody responses targeting different antigenic sites in pigs vaccinated with FMDV serotypes A and O. (A) Archived sera from ten pigs immunized twice with the A/WH/CHA/09 vaccine were collected at 56 days post-initial immunization (dpi) and tested for competition with neutralizing monoclonal antibodies pOA-20 (Site 1), W125 (Site 2), W2 (Site 4), and W145 (Site 5). (B) Archived sera from ten pigs immunized twice with the O/HN/CHA/93 vaccine were collected at 56 dpi and tested for competition with neutralizing monoclonal antibodies pOA-20 (VP1 G-H loop), F145 (Site 2), C4 (Site 4), and B66 (VP1 C-terminus). Statistical analysis was performed using One-Way ANOVA followed by a multiple-comparison test with a 95% confidence interval in GraphPad Prism 8.0. ns indicates no significant difference. *, **, *** indicate significant differences at $P < 0.05$, $P < 0.01$, $P < 0.001$, respectively. Table A. The interaction residues of FMDV Asia1/JS/05 with PAS5. Table B. Interface identification and interaction analysis of PAS5 with FMDV Asia1/JS/05 by the PISA Program. Table C. The interaction residues of FMDV Asia1/JS/05 with PAS12. Table D. Interface identification and interaction analysis of PAS12 with FMDV Asia1/JS/05 by the PISA Program. Table E. Summary of FMDV-neutralizing monoclonal antibodies and their recognized antigenic sites. Table F. Cryo-EM data collection and refinement statistics. (DOCX)

## Acknowledgments

We thank the Supercomputing Center of Lanzhou University for providing facilities and the staff at Instrument Centre, Lanzhou Veterinary Research Institute, Chinese Academy of Agricultural Sciences for excellent assistance in single cell sorting using BD FACS Aria II. Illustrations in Figs 1A and 7D were created with BioRender.com, with thanks to Y. Chen for her assistance in visual presentation.

## Author contributions

**Conceptualization:** Shulun Huang, Kun Li, Zengjun Lu, Dongsheng Lei, Qiang Zhang.

**Data curation:** Shulun Huang, Shanquan Wu, Fengjuan Li, Jiaxin Yang, Jian Wang.

**Formal analysis:** Shulun Huang, Shanquan Wu, Fengjuan Li, Kaiheng Dong.

**Funding acquisition:** Yimei Cao, Kun Li, Zengjun Lu, Dongsheng Lei.

**Investigation:** Shulun Huang, Shanquan Wu, Fengjuan Li, Kaiheng Dong, Jiaxin Yang, Hehe Zhang.

**Methodology:** Shulun Huang, Fengjuan Li, Pinghua Li, Pu Sun, Yimei Cao, Huifang Bao, Hong Yuan, Zaixin Liu.

**Resources:** Pinghua Li, Pu Sun, Yong Peng, Dongsheng Lei, Qiang Zhang.

**Software:** Pinghua Li, Qiongqiong Zhao, Xingwen Bai, Jinlian Hua, Dongsheng Lei.

**Supervision:** Jinlian Hua, Zengjun Lu, Qiang Zhang.

**Validation:** Shulun Huang, Shanquan Wu, Huifang Bao, Dong Li, Yuanfang Fu, Jing Zhang.

**Visualization:** Shulun Huang, Shanquan Wu, Hehe Zhang, Ying Sun, Xueqing Ma, Zhixun Zhao.

**Writing – original draft:** Shulun Huang, Shanquan Wu.

**Writing – review & editing:** Shulun Huang, Kun Li, Jinlian Hua, Zengjun Lu, Qiang Zhang.

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
