## [Decision Letter · Decision Letter 0]

3 Oct 2025

Porcine B cell receptor repertoire uncovers immunodominant properties of antigenic structures on serotype Asia1 foot-and-mouth disease virus

PLOS Pathogens

Dear Dr. Lu,

Thank you for submitting your manuscript to PLOS Pathogens. After careful consideration, we feel that it has merit but does not fully meet PLOS Pathogens's publication criteria as it currently stands. Therefore, we invite you to submit a revised version of the manuscript that addresses the points raised during the review process.

Please submit your revised manuscript within 60 days Dec 02 2025 11:59PM. If you will need more time than this to complete your revisions, please reply to this message or contact the journal office at plospathogens@plos.org. Please include the following items when submitting your revised manuscript:

We look forward to receiving your revised manuscript.

Kind regards,

Sonja M. Best, Ph.D.

Section Editor

PLOS Pathogens

Sonja Best

Section Editor

PLOS Pathogens

Editor-in-Chief

PLOS Pathogens

orcid.org/0000-0003-2946-9497

Editor-in-Chief

PLOS Pathogens

orcid.org/0000-0002-7699-2064

**Journal Requirements:**

At this stage, the following Authors/Authors require contributions: Shulun Huang, Shanquan Wu, Fengjuan Li, Pinghua Li, Pu Sun, Yimei Cao, Huifang Bao, Kaiheng Dong, Jiaxin Yang, Qiongqiong Zhao, Ying Sun, Dong Li, Xingwen Bai, Yuanfang Fu, Hong Yuan, Xueqing Ma, Zhixun Zhao, Jing Zhang, Jian Wang, Zaixin Liu, Yong Peng, Kun Li, Jinlian Hua, Zengjun Lu, Dongsheng Lei, and Qiang Zhang. Please ensure that the full contributions of each author are acknowledged in the "Add/Edit/Remove Authors" section of our submission form.

Potential Copyright Issues:

- Figures 1 and 7. Please confirm whether you drew the images / clip-art within the figure panels by hand. If you did not draw the images, please provide (a) a link to the source of the images or icons and their license / terms of use; or (b) written permission from the copyright holder to publish the images or icons under our CC BY 4.0 license. Alternatively, you may replace the images with open source alternatives. See these open source resources you may use to replace images / clip-art:

5) Please ensure that the funders and grant numbers match between the Financial Disclosure field and the Funding Information tab in your submission form. Note that the funders must be provided in the same order in both places as well.

State what role the funders took in the study. If the funders had no role in your study, please state: "The funders had no role in study design, data collection and analysis, decision to publish, or preparation of the manuscript.".

**Reviewers' Comments:**

Reviewer's Responses to Questions

**Part I - Summary**

Reviewer #1: In this paper, the authors analyze the BCR repertoire of a pig immunized with and FMDV Asia1 inactivated vaccine using B-cell immune separation, expressed the mAbs in 293F cells, and determined that 17 mAbs neutralized FMDV Asia1. Isolation and sequence analysis of neutralization escape mutants revealed 3 groups: 1) 14 mAbs mapped to the VP2 B-C loop corresponding to antigenic site 2, 2) 2 mAbs mapped to the VP3 B-B knob and VP2 B-C loop corresponding to a novel antigenic site composed of site 2 and 4, and 3) 1 mAb mapped to the C-terminus of VP1. Cryo-EM analysis of mAb PAS12 form group 2 confirmed the identification of the novel epitope antigenic structure composed by VP2 and VP3 residues. The authors concluded that the VP2 B-C loop is the immunodominant epitope in FMDV Asia1. This is an interesting work that further analyzed the antibody response against FMDV Asia1 in pigs, which has not been studied in detail.

Reviewer #2: The manuscript by Huang and colleagues addresses an interesting and important topic. The application of porcine B cell receptor repertoire sequencing to investigate antigenic structures of FMDV serotype Asia1 is relatively novel and has potential implications for vaccine development. However, there are several critical areas where the manuscript could be improved.

Reviewer #3: Overall this is an excellent and highly valuable study that adds to our limited understanding of natural host responses to vaccine induced FMD protection. The methods are established but remain contemporary and are used to answer some very specific questions, trying to better determine the antigenic landscape of a viral capsid that has been used so effectively to eradicate the Asia1 serotype from China. Taking a comparative approach to epitope availability compared to other serotypes that remain a threat is a powerful approach and a potentially useful approach.

A clear strength of the work is the high resolution methodologies that have been used effectively to look at the molecular interactions and mechanisms of virus neutralisation. However, the overall conclusions that are drawn rely on one young pig being representative of all pigs, and in think this needs to be tempered throughout the manuscript. The clear phenotyping of antigen specific B cells and the unusual role of the light chain are very interesting and novel findings that help us to better understand protective responses in pigs, not only to FMDV.

**Part II – Major Issues: Key Experiments Required for Acceptance**

Reviewer #1: The paper requires editing of the English language as well as technical editing because it is difficult to understand the ambiguous meaning of many sentences. It is surprising that no mAbs were isolated against the VP1 140-142 immunodominant neutralizing epitope that contains the RGD receptor binding site. The authors attributed this unexpected issue to the unique structure of the FMDV Asia1 particles that differs from other FMDV serotypes and to the distinctive porcine light-kappa antibody response, explanations that are not well substantiated or convincing enough to this reviewer. Furthermore, the authors did evaluate some study design pitfalls such as the effect of the inactivation process in the antigenic structure of FMDV Asia1 or the use of only one pig in the BCR repertoire study.

Reviewer #2: 1. The manuscript, including the abstract, contains frequent grammatical and syntactic issues that affect readability and clarity. Common problems include incorrect verb tenses, subject-verb agreement errors, awkward phrasing, and incomplete clauses. Even in the abstract, there are multiple issues. For example, a sentence begins with “while” but is not followed by a main clause. These problems make the manuscript difficult and at times unpleasant to read. A thorough language revision is strongly recommended before further consideration.

2. The use of terminology, particularly around the concept of immunodominance, is inconsistent and at times confusing. Phrases such as “equal immunodominances” and “average immunodominance” are not standard and are grammatically incorrect. The term "immunodominance" is typically treated as an uncountable noun and should not be pluralized. The authors are encouraged to revise the language to reflect more conventional and precise scientific usage. Terms like “even immunodominance” or “evenly distributed immunodominance” would be more appropriate and easier to interpret.

3. A significant limitation of the study is that the B cell repertoire analysis and monoclonal antibody discovery were based on a single animal (pig #2219). Although sera from 10 pigs were used for follow-up serological testing, the central sequencing and epitope mapping results are derived from just one individual. Since pigs are an outbred species, there is likely to be substantial variation in immune gene usage and antibody responses across individuals. This raises concerns about the generalizability of the findings. The authors should clearly acknowledge this limitation and, ideally, include additional pigs in the BCR analysis to assess the reproducibility of the observations.

4. The number and diversity of neutralizing antibodies identified also seem limited. Of the 33 monoclonal antibodies expressed, only 17 were neutralizing, and many of these shared similar VH and VL gene segments, particularly IGHV1-4 and IGKV1-11. This redundancy calls into question how representative the antibody panel is and whether it captures the full range of epitope diversity. The authors should address whether this pattern reflects a biological feature of the Asia1-specific response or is a consequence of sampling bias. They should also consider whether the current dataset is sufficient to support broad conclusions about immunodominance.

5. The manuscript suggests that a more evenly distributed pattern of immunodominance may contribute to the successful eradication of FMDV serotype Asia1. While this is a compelling idea, there are no comparative data from serotype O or A included under similar experimental conditions. Without such controls, it is difficult to evaluate whether the observed response is truly unique to Asia1. The inclusion of cELISA or BCR repertoire data from pigs immunized with serotype O or A would provide essential context and help strengthen the manuscript’s central claim.

6. The competitive ELISA presented in Figures 7F and 7G is central to the conclusion regarding evenly distributed immunodominance, yet the methodology for this assay is not described. Critical details are missing, including how the assay was designed, how the serum samples were diluted, and how competition was measured. As this assay supports one of the manuscript’s main conclusions, a complete and detailed description of the protocol should be included.

Reviewer #3: (No Response)

**Part III – Minor Issues: Editorial and Data Presentation Modifications**

Reviewer #1: 1) Lines 117-120. This sentence needs to be edited. What the authors probably mean is that they purified IgM- B cells by negative magnetic separation.

2) Line 204. Binding affinities of all mAbs were not measured in this study so it cannot be determined if they are different. Please, change or delete this sentence.

3) Lines 233-238, Fig 3C. The ELISA procedure should be moved to the M&M section.

4) Lines 246-252. Unclear concept that should be move to the discussion section or deleted.

5) Lines 423-425 and Figures 7F-G. it is unclear how the cELISA was performed. There is no description in M&M for this experiment. What are the positive and negative controls?

6) Lines 448-458. The experiments in Figs 7F-G do not establish "equal immunodominance" of the epitopes but rather seem to indicate that the mAbs compete to some extent with the polyclonal antibodies from vaccinated pigs. It is unclear what is the meaning and relevance of these experiments as performed.

7) Discussion. The authors showed in this paper that under their experimental conditions a single immunized pig with inactivated FMDV Asia1 vaccine developed an unexpected antibody response directed not against the immunodominant VP1 140-142 site 1 epitope, which contains the RGD receptor-binding motif, but rather skewed towards site 2 and 4 epitopes. However, the authors in a previous publication (Li et al., 2024) showed that pigs immunized with FMDV serotypes O and A develop anti- VP1 140-142 antibodies that cross-neutralize serotype Asia1. Also immunized mice against FMDV Asia1 develop neutralizing antibodies against VP1 140-142 (Grazioli et al., 2013). A better discussion of these somehow contradictory results will increase the relevance of this paper. The authors also failed to discuss many other relevant issues such as how the genetic background and BCR repertoire of the only pig used in the single-cell sequencing analysis and production of mAbs may have skewed their results.

8) How was the FMDV used in the vaccination inactivated, and what tests were performed to verify the antigenic integrity of the inactivated virus.

Reviewer #2: (No Response)

Reviewer #3: General comments that I think would improve the manuscript.

The language throughout the document needs editing to make the text clearer and grammatically accurate.

The text in the figures was usually too small in the reviewers PDF version.

The figure legends would benefit from revision and more specific language throughout.

Pigs have a relatively unique pathology after FMDV infection compared to ruminants. Some information in the introduction related to

Comments that should be addressed prior to publication.

1. Can you be more explicit about the selection criterion applied to the antibodies to selected candidates for cryoEM? For example, the antibody that binds the C terminus of VP1 (PAS32) was not pursued for structural analysis.

2. There is a clear difference based on epitope dominance between the monoclonal antibodies and the serum/competitive ELISA. It should be noted that the vaccinated animals were older than the piglet used for B cell isolation and it would be useful to understand if there was no gross difference in the overall response of these animals, repertoire development etc. The equal dominance of the antigenic sites, after boost perhaps suggests that there is dominance during the first encounter? Without avidity measurements for the monoclonal antibodies, the competitive ELISA from one animal is very hard to interpret more broadly. Taken together, I think the overall conclusion needs to be modified to reflect this is a pilot approach and may not be representative.

3. As a clear purpose of the paper is to identify epitopes that could support vaccine design across serotypes, the comparative analysis between the identified epitopes and the key serotypes that remain a threat is lacking. This may be an alignment or some in silico modelling approach, I am not proposing any more animal work. I think this would greatly improve the overall significance of the work and could better support the claims around immunogenicity differences and how this work could lead to better vaccines.

4. The statement in the abstract and repeated in other parts of the manuscript stating that this knowledge "may facilitate the selection of a broader range of germline gene segments and enhance antibody diversity production, potentially conferring exceptional immunogenicity of Asia1 vaccine for viral eradication" is not linked to the findings in the manuscript or the discussion. It would benefit the paper if the authors were more explicit about this potential based on the antibodies that they eventually identified.

5. Line 192. Clustering B cells based on 100% identity of the heavy chain is very strict when you state clearly that the IgG compartment has undergone significant mutation and will no doubt be maturing. This may in part explain why the number of epitopes that you identified from the relatively middle to low frequency clusters is limited and it does not appear that you have studied any non-neutralising binding antibodies. From Table 3D it would appear that half of your antibodies are non-neutralising? Considering the above, do you think it is correct to say that this is the repertoire of antigen specific B cells as I am not sure evidence is provided, but assumed from the sorting. In addition, I therefore think the epitopes identified are not exhaustive. The data is excellent, the interpretation needs to be refined.

PLOS authors have the option to publish the peer review history of their article (what does this mean? ). If published, this will include your full peer review and any attached files.

**Do you want your identity to be public for this peer review?** For information about this choice, including consent withdrawal, please see our Privacy Policy .

Reviewer #1: No

Reviewer #2: No

Reviewer #3: No

**Figure resubmission:**

**Reproducibility:**



---

## [Decision Letter · Decision Letter 1]

10 Dec 2025

PPATHOGENS-D-25-00403R1

Porcine B cell receptor repertoire uncovers immunodominant properties of antigenic structures on serotype Asia1 foot-and-mouth disease virus

PLOS Pathogens

Dear Dr. Lu,

Thank you for submitting your revised manuscript to PLOS Pathogens. After careful consideration, we and reviewers feel that it has merit but does not fully meet PLOS Pathogens's publication criteria as it currently stands. Therefore, we invite you to submit another revised version of the manuscript that addresses the points raised during the review process with contributions of two original and one new reviewers.

We look forward to receiving your revised manuscript.

Kind regards,

Alexander E. Gorbalenya, Ph.D., D.Sci.

Section Editor

PLOS Pathogens

Sumita Bhaduri-McIntosh

Editor-in-Chief

PLOS Pathogens

orcid.org/0000-0003-2946-9497

Michael Malim

Editor-in-Chief

PLOS Pathogens

orcid.org/0000-0002-7699-2064

**Additional Editor Comments:**

Please check if the title may be improved after two rounds of the manuscript revision.

**Reviewers' Comments:**

Reviewer's Responses to Questions

**Part I - Summary**

Reviewer #2: The authors have addressed my previous comments, although not as fully as I had hoped. Overall, the revisions are acceptable. All reviewers raised concerns about the representativeness of a BCR repertoire generated from a single animal, and the authors now acknowledge this limitation in the Discussion, which is appropriate.

In their response to Question 21, the authors state that "this study represents a pilot exploration of antigen-specific B cell responses in pigs." However, the manuscript itself does not clearly present the work as a pilot study. I strongly recommend that the authors explicitly include the word "pilot" or "preliminary" in the Discussion or even in the Abstract. This would more accurately reflect the scope of the study.

Although I believe the authors' conclusions are likely correct, the reliance on a single pig for BCR sequencing means that the results could differ if a different animal were used. Emphasizing that this is a preliminary study will help ensure that readers interpret the findings with the proper context and caution.

Reviewer #3: The authors have addressed all the comments well, which does directly address the limitations of the study.

Reviewer #4: The studies by Shun Huang et al., described in this revised manuscript have analysed the production of neutralising antibodies in pigs against a foot-and-mouth disease virus (FMDV) vaccine, serotype Asia1. The authors characterised the repertoire of antibodies produced from a single pig and identified the features of the virus that these antibodies recognise. These sites were largely shared with those identified previously in other studies on FMDV but it is useful to have this information for pigs. The authors suggest that the sites vary in their dominance between that observed with other serotypes. It is proposed that these results may assist in rational vaccine design.

**Part II – Major Issues: Key Experiments Required for Acceptance**

Reviewer #2: (No Response)

Reviewer #3: No

Reviewer #4: 1) It is regrettable that the antibody repertoire is only determined from a single pig, it is not, therefore, certain that the immune responses from this animal are representative of the pig population, the study would be greatly strengthened by analysing one or more further, unrelated, animals. The authors do indicate that this is a limitation of their study, after the comments from other reviewers.

2) The serological responses of 10 pigs to the Asia1 vaccine was assessed in a competition ELISA. However, it may be that the process of coating the FMDV antigen onto ELISA plates may change its surface properties and perhaps some sites may be less well represented than in the native 146S particles. This may make the assessment of site dominance problematic. The authors should at least comment on this. I think it would be better if the discussion of the relative dominance of these sites was much reduced.

3) The authors cite reference 4 for the eradication of Asia1 FMDV in China but the reference is actually focussed on serotypes O and A, an alternative citation should be used. It is important to know whether the successful eradication of Asia1 in China included widespread vaccination of both pigs and cattle. This has implications for the utility of understanding the immune response repertoire to FMDV antigens in pigs.

**Part III – Minor Issues: Editorial and Data Presentation Modifications**

Reviewer #2: (No Response)

Reviewer #3: No

Reviewer #4: a) The use of English is still poor in places although the sense can usually be followed.

b) Line 202, the text refers to “Asia1-specific BCR”, as far as I understand it, the methodology does not really demonstrate specificity for Asia!, this was just the reagent used in the screening. Perhaps some of the BCRs would have also recognised other FMDVs since cross-serotypic antibodies have been identified previously.

c) Lines 314-315 (and maybe elsewhere). Mutations occur in nucleic acids not in amino acid sequences, the changes in protein sequence should be referred to as substitutions.

PLOS authors have the option to publish the peer review history of their article (what does this mean? ). If published, this will include your full peer review and any attached files.

**Do you want your identity to be public for this peer review?** For information about this choice, including consent withdrawal, please see our Privacy Policy .

Reviewer #2: No

Reviewer #3: No

Reviewer #4: No

**Figure resubmission:**
---

## [Editor Report · Decision Letter 2]

12 Jan 2026

Dear Dr. Lu,

We are pleased to inform you that your manuscript 'Porcine B cell receptor repertoire uncovers balanced recognition of antigenic structures on serotype Asia1 foot-and-mouth disease virus' has been provisionally accepted for publication in PLOS Pathogens.

Best regards,

Alexander E. Gorbalenya, Ph.D., D.Sci.

Section Editor

PLOS Pathogens

Sumita Bhaduri-McIntosh

Editor-in-Chief

PLOS Pathogens

orcid.org/0000-0003-2946-9497

Michael Malim

Editor-in-Chief

PLOS Pathogens

orcid.org/0000-0002-7699-2064
---

## [Editor Report · Acceptance letter]

Dear Dr. Lu,

We are delighted to inform you that your manuscript, "Porcine B cell receptor repertoire uncovers balanced recognition of antigenic structures on serotype Asia1 foot-and-mouth disease virus," has been formally accepted for publication in PLOS Pathogens.

Best regards,

Sumita Bhaduri-McIntosh

Editor-in-Chief

PLOS Pathogens

orcid.org/0000-0003-2946-9497

Michael Malim

Editor-in-Chief

PLOS Pathogens

orcid.org/0000-0002-7699-2064